# PARP7 is a proteotoxic stress sensor that labels proteins for degradation

Nonso J Ikenga ⬤, Jörg Vervoorts ⬤, Bernhard Lüscher ⬤, Roko Žaja ⬤ ✉ & Karla L H Feijs-Žaja ⬤ ✉

## Abstract

**ADP-riboslylation is a post-translational modification that plays a critical role in cellular stress responses. We have observed that during proteotoxic stress, cellular ADP-ribosylation increases, with ADP-ribosylated proteins accumulating in cytoplasmic foci containing ubiquitin and p62. During prolonged stress, these ADP-ribosylated proteins are transported to aggresomes and subsequently degraded via autophagy. In the absence of ubiquitination, ADP-ribosylated proteins become more prevalent and less soluble, indicating that ubiquitination is indispensable for this process. Upon inhibition of PARP7, accumulation of mono(ADP-ribosyl)ated proteins in response to proteotoxic stress is impeded. PARP7 turnover is very high under normal conditions; however, the protein becomes stabilised following proteotoxic stress and thereby forms an ideal proteotoxic stress sensor. Our findings imply that, contrary to the current paradigm, not all ADP-ribosylation may occur on specific sites to regulate specific protein characteristics. Instead, it may be rather promiscuous to enable efficient protein degradation or segregation to prevent irreversible damage caused by defective proteins.**

**Keywords** Protein Degradation; ADP-ribosylation; PARP; Macrodomain; Ubiquitination
**Subject Categories** Autophagy & Cell Death; Post-translational Modifications & Proteolysis

## Introduction

ADP-ribosylation is a post-translational modification that results in the transfer of ADP-ribose (ADPr) from nicotinamide adenine dinucleotide (NAD⁺) to a range of amino acids, including glutamates, serines and cysteines. The intracellular enzymes that catalyse this reaction are ADP-ribosyltransferases (ARTs), best known as poly(ADP-ribose) polymerase (PARP) and TNKS proteins (Hottiger et al, 2010; Luscher et al, 2022). The 17 PARPs can be divided into two major categories: 4 PARPs that catalyse the formation of polymers of ADP-ribose, poly(ADP-ribosyl)ation (PARylation) and 12 PARPs that transfer a single ADPr, mono(ADP-ribosyl)ation (MARylation) (Challa et al, 2021; Gibson and Kraus, 2012; Hottiger et al, 2010; Luscher et al, 2022; Schreiber et al, 2006). For the remaining PARP, PARP13, no catalytic activity has been measured to date. Both PARylation and MARylation are fully reversible processes, catalysed by hydrolases from two structurally distinct protein families, the macrodomain-containing proteins (MDCPs) and the ADP-ribosylhydrolases (ARHs) (Feijs et al, 2013a; Rack et al, 2018; Rack et al, 2020). PARylation by PARP1 has, for example, a well-studied role in DNA damage repair (Alemasova and Lavrik, 2019; Azarm and Smith, 2020; Ray Chaudhuri and Nussenzweig, 2017), whereas PARylation generated by TNKS1 can lead to recruitment of specific E3 ubiquitin ligases via their PAR-binding domains (Callow et al, 2011; He et al, 2012; Huang et al, 2009; Perrard and Smith, 2023; Zhang et al, 2011). These subsequently poly-ubiquitinate their targets, leading to the substrates' proteasomal degradation.

In contrast to PARylation, much less is known about MARylation. Past studies have attempted to identify monoART substrates with in vitro screening approaches (Feijs et al, 2013b), and more recent work has focused on mass spectrometry (MS) of cells with overexpressed monoARTs (Carter-O'Connell et al, 2016; Carter-O'Connell et al, 2018; Palavalli Parsons et al, 2021; Rodriguez et al, 2021; Rodriguez and Cohen, 2022). Collectively, these studies have identified thousands of potentially MARylated proteins in diverse organisms, cells and tissue types. Despite these efforts, relatively little is still known about the function of MARylation, especially when it comes to studying the consequences of modification for single-substrate proteins. Intriguingly, for several PARP7 substrates it has been described that MARylation influences their turnover, as MARylation, for example, induces degradation of hypoxia-inducible factor alpha (HIF1a) and aryl hydrocarbon receptor (AHR) (Rijo et al, 2021; Zhang et al, 2020). In contrast, FRA1 is stabilised upon PARP7-dependent MARylation (Manetsch et al, 2023). In addition to PARP7, also PARP14 and PARP9/DELTEX E3 may play a role in protein turnover as PARP14, ADPr, ubiquitin and p62 (also known as SQSTM1) colocalise in cytoplasmic condensates in response to interferon gamma stimulation of cells (Kar et al, 2024; Kubon et al, 2024; Raja et al, 2025; Ribeiro et al, 2024). Although the function of these foci is not clear, PARP14 protein levels were shown to be regulated in an ADPr-dependent manner. Also, PARP10 co-localises with p62 and ubiquitin with thus far unknown function (Kleine et al, 2012). A last piece of evidence for a possible role of ADP-ribosylation in regulation of protein turnover, derives from observations that ubiquitin can be

Institute of Biochemistry and Molecular Biology, RWTH Aachen University, Aachen, Germany. ✉E-mail: rzaja@ukaachen.de; kfeijs@ukaachen.de

modified by MARylation (Ahmed et al, 2020; Chatrin et al, 2020; Yang et al, 2017) and vice versa, ADP-ribose can be ubiquitinated (Bejan et al, 2025; Zhu et al, 2022). Although not investigated yet, it is thinkable that modification of ubiquitin will influence the turnover of modified proteins.

Inhibition of the proteasome by MG132 leads to an increased ADP-ribosylation signal when PARP6, PARP7, PARP12 or PARP15 are overexpressed (Weixler et al, 2023). Whether the resulting proteotoxic stress activates the respective PARPs to increase cellular ADP-ribosylation, or whether those PARPs are stabilised by MG132 and therefore the ADP-ribosylation signal appears stronger, is unclear. In the current work, we have asked the question of whether there might be a generic role for protein MARylation in the cellular response to proteotoxic stress. We observe that cytoplasmic foci form that are characterised by containing p62, ubiquitin, and ADPr. Interfering with proteasomal degradation results in aggresome formation, which contains MARylated substrates. These are then transferred to autophago-somes, suggesting degradation by autophagy. Thus, MARylation serves as a degradation signal in response to stress.

# Results

## MARylated proteins localise to cytoplasmic foci upon perturbation of cellular ubiquitination

Under basal conditions, MARylation levels are very low, and it has therefore been suggested that the modification of proteins with a single ADP-ribose occurs rarely in basal conditions, but rather is induced in response to cellular stress (Feijs and Zaja, 2022; Weixler et al, 2023). To test whether protein MARylation may play a role in protein degradation similar to the described role for PARylation, we decided to make use of recently developed detection reagents for MARylation (Bonfiglio et al, 2020; Gibson et al, 2017; Hopp et al, 2021; Nowak et al, 2020; Weixler et al, 2023) and analysed where in the cells ADP-ribosylation occurs. Fixation of cells in PFA results in a strong mitochondrial staining using a MAR/PAR antibody, which probably results from mitochondrial NADH that is cross-linked and detected by the MAR/PAR antibody (Weixler et al, 2023). For this reason, we fixed the cells using methanol instead. Methanol fixation resulted in a weak diffused staining of ADP-ribosylated proteins in the untreated cells (Fig. 1A). To determine whether MARylation may be involved in the cellular response to proteotoxic stress, we used a general inhibitor of deubiquitinases (DUBs), PR-619, to increase the amount of ubiquitinated proteins and thereby perturb protein homoeostasis (Altun et al, 2011). Upon 30 min treatment of HeLa cells with PR-619 (named DUBi in the figures) we observed an accumulation of ADP-ribosylated proteins in defined cytoplasmic foci (Fig. 1A). To exclude that the staining is an artefact of methanol fixation we also used glyoxal as a fixative. Glyoxal is similar to PFA a cross-linking agent but with shorter cross-linking arms (Richter et al, 2018). Using glyoxal, we could confirm the accumulation of ADP-ribosylated proteins in cyto-plasmic foci upon PR-619 treatment as seen in methanol fixed cells (Fig. 1B). This confirmed our observation that MARylated proteins accumulate in cytoplasmic foci upon inhibition of deubiquitination.

The MAR/PAR antibody used in these experiments recognises the adenine moiety of ADPr, and it therefore detects both mono(ADP-ribose) (MAR) as well as poly(ADP-ribose) (PAR) (Weixler et al, 2023). To determine whether the observed signal is caused by MAR- or PARylated proteins, or even free PAR, we repeated the staining with recently developed MARylation-specific antibodies (Bonfiglio et al, 2020). Both methanol and glyoxal-fixed cells showed the presence of cytoplasmic foci as observed using the MAR/PAR antibody, confirming that these foci contain MARylated proteins (Fig. 1C,D). Using a MAR-specific antibody, we do not observe nuclear staining (Fig. 1D), implying that the nuclear staining seen with a PAR/MAR antibody in untreated cells fixed using glyoxal (Fig. 1B) is most likely PAR.

## MARylated proteins co-localise with ubiquitin and p62 upon proteotoxic stress in p62 bodies

Incubation of cells with the DUB inhibitor PR-619 leads to an accumulation of ubiquitinated proteins, resulting in proteome imbalance and proteotoxic stress due to overload of the proteasomal degradation system. Under these conditions, p62 interacts with ubiquitinated proteins and promotes the formation of aggresome-like induced structures (referred to as ALIS or p62 bodies) (Johnston and Samant, 2021). To further characterise the observed MAR foci and to determine whether MARylated proteins accumulate in p62 bodies, we co-stained MARylated proteins with ubiquitin and p62. Upon DUB inhibitor treatment, MARylated proteins co-localised with both ubiquitin and p62 in cytoplasmic foci (Fig. 2A,B). p62 bodies are also induced in response to proteasomal overload or inhibition. When the ubiquitin-proteasome system is dysfunctional, p62 bodies represent the first step in the process of protein degradation by selective autophagy (Bjorkoy et al, 2005; Danieli and Martens, 2018; Johnston and Samant, 2021). We observed co-localisation of MARylated proteins with both p62 and ubiquitin after inhibiting the proteasome with a short-term treatment of the proteasome inhibitor bortezomib (BTZ), indicating that MARylated proteins accumulate in p62 bodies during proteasomal dysfunction (Fig. 2C,D). The detectable presence of ADP-ribosylated proteins in p62 bodies can have different causes: either these proteins are usually dispersed throughout the cell and become visible only upon local concentra-tion during stress, or the levels of ADP-ribosylated proteins increase in this condition. Using western blotting, we observe an increase in overall ADP-ribosylation levels after proteasome inhibition in two different cell lines, implying that ADP-ribosylation is induced in this condition (Fig. 2E). When we performed a time course with bortezomib, we noticed that in both HeLa and U2OS cells protein ubiquitination is stabilised immedi-ately after inhibiting the proteasome, followed by a markedly slower accumulation of ADP-ribosylation (Fig. 2F). This implies that ADPr is not a mark which occurs continuously to regulate protein turnover, as are certain ubiquitin species, but rather is induced in response to proteotoxic stress. The most prominent signal in HeLa cells is present at around 75 kDa, which is also observed in U2OS, albeit with lesser intensity and not responding to the inhibitor treatment. In U2OS cells, the most prominent ADP-ribosylation induced by proteasome inhibition is visible at a larger size. As PARP7 and PARP1 are approximately 75 and 115 kD, respectively, it is possible that the strongest signals reflect automodified PARPs. This could imply that different transferases or hydrolases are active in these cells, in line with, for example, differences in PARG activity

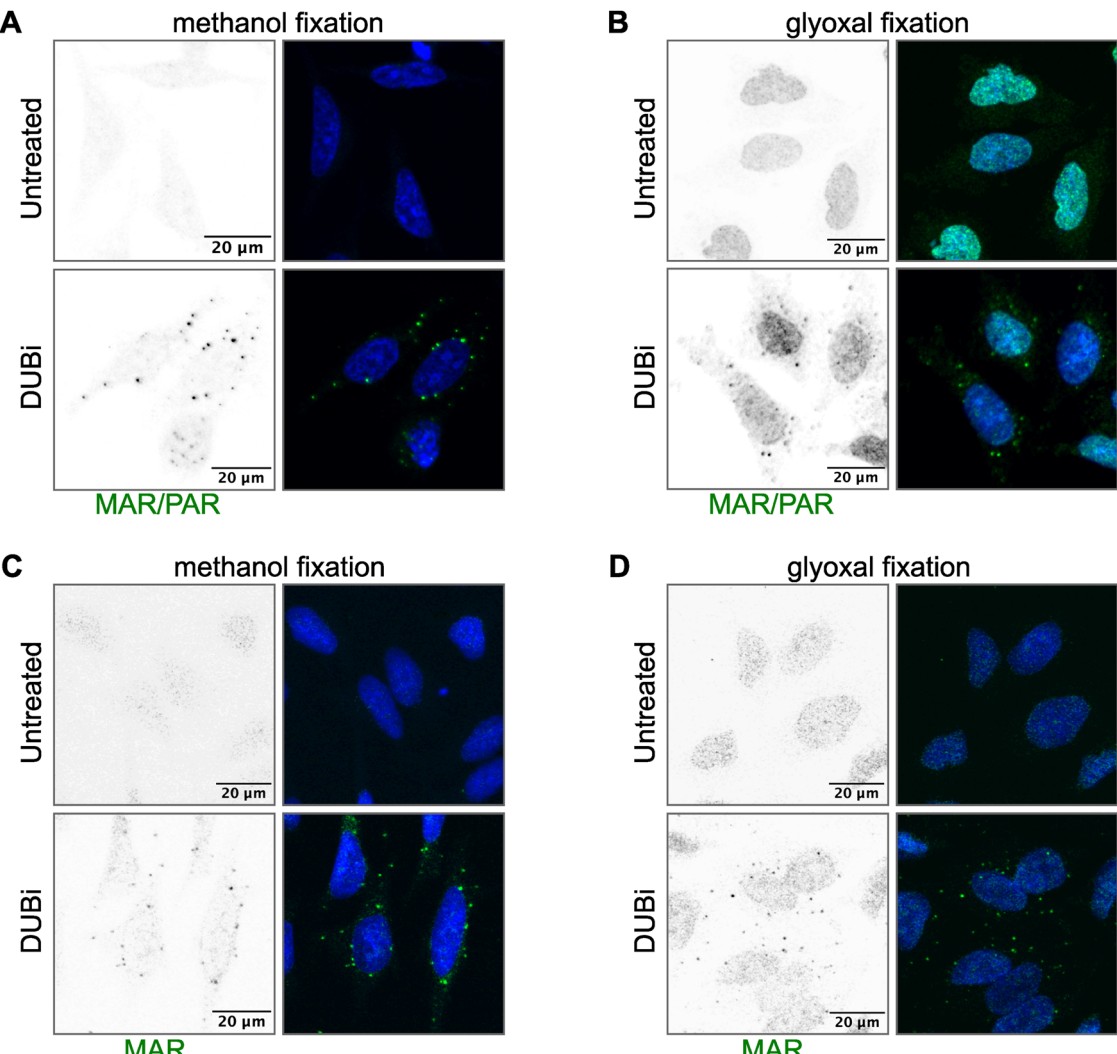

**Figure 1.  ADP-ribosylated proteins accumulate in cytoplasmic foci upon perturbation of protein homoeostasis.**

(A) HeLa cells were treated with 10 µM deubiquitinase inhibitor (DUBi) PR-619 for 30 min, followed by fixation in methanol and staining using a PAR/MAR antibody from Cell Signaling Technology (E6F6A). The mounted cells were imaged using a confocal microscope. (B) As in (A), with fixation of the cells in glyoxal. (C, D) As in (A, B), using a mono(ADP-ribose)-specific antibody from BioRad (HCA354). Scale bars represent 20 µm. To enhance the visibility of foci, grayscale images of the MAR/PAR or MAR signals were inverted and shown on the left of each panel. On the right of each panel, merged images displaying DNA in blue and MAR/PAR or MAR signal in green.

levels we have observed previously between HEK293 and HeLa cells (Weixler et al, 2025).

## MARylated proteins in p62 bodies are transported to aggresomes and degraded through autophagy

To determine the fate of MARylated proteins in p62 bodies, we inhibited proteasomal degradation for a longer period with a proteasome inhibitor and asked where the MARylated proteins are localised. In addition to localisation of ADP-ribosylated proteins to p62 bodies, which arise soon after proteasome inhibition, we now observe ADP-ribosylation in large, perinuclear structures that are surrounded by vimentin (Fig. 3A). This corresponds to the typical appearance of aggresomes, which corresponds to the structures where proteins from p62 bodies are actively transported to, and

from where they are further destined to degradation via the autophagic machinery (Johnston and Samant, 2021). This indicates that ADP-ribosylated proteins can indeed be further processed from p62 bodies to aggresomes. To test whether these proteins are indeed transferred to autophagosomes, we next interfered with both proteasomal degradation as well as autophagy. An increased number of ADPr foci could be observed when both proteasome and autophagic degradation pathways were inhibited during a 4-h treatment period (Fig. 3B,C). The increased number of ADPr foci upon blockage of autophagy implies that autophagy is required for the turnover of these foci. Using western blotting of both soluble and insoluble material, we observed that the increase in ADP-ribosylation following MG132 treatment can be slightly increased by blocking autophagy using bafilomycin A, most pronounced in the RIPA-insoluble fraction (Fig. 3D). To ensure that this is not an

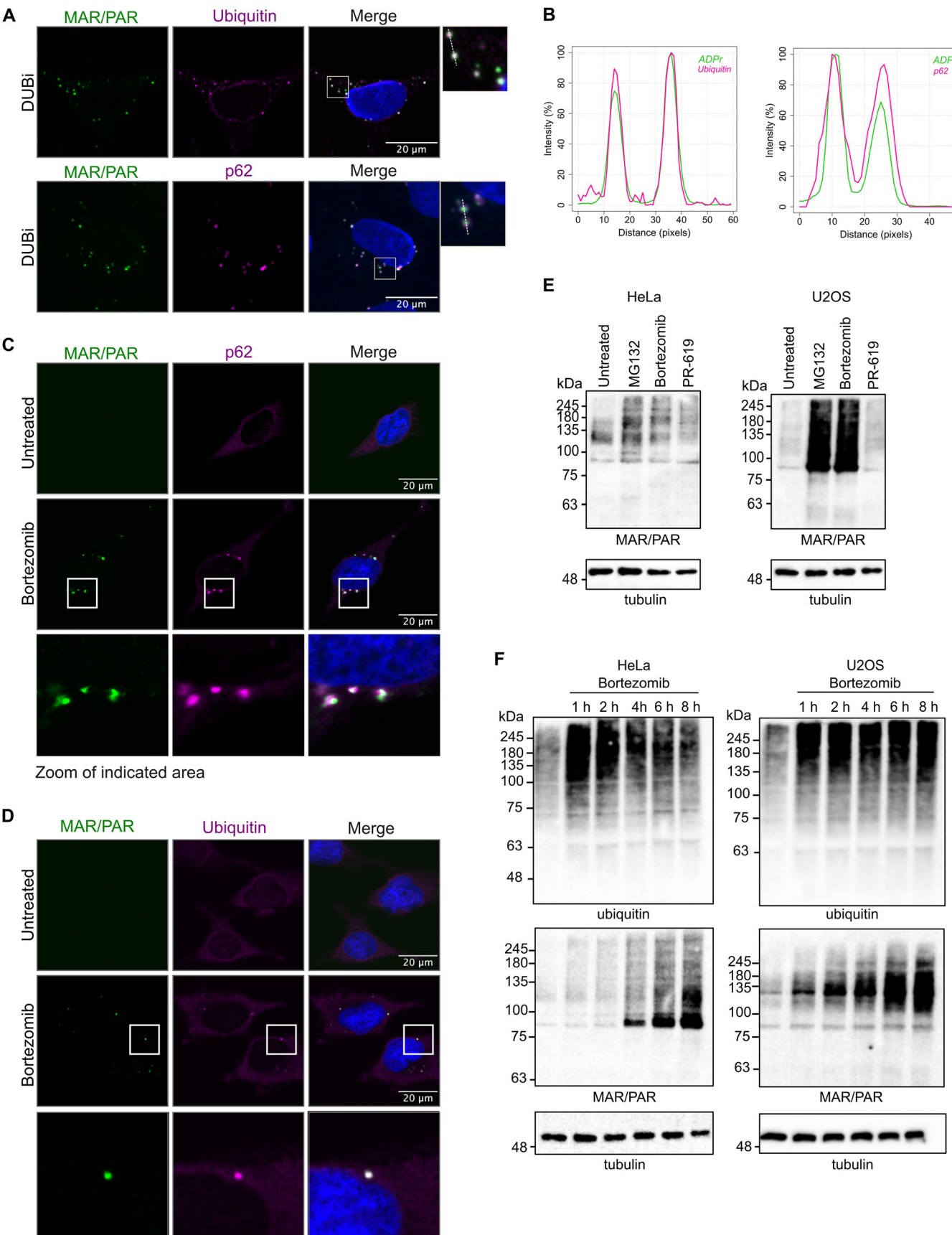

Figure 2.  Interference with protein homoeostasis leads to accumulation of ADP-ribosylated proteins.

(A) HeLa cells were treated with 10 μM PR-619 (DUBi) for 30 min, fixed in methanol and stained with antibodies for MAR/PAR and ubiquitin (top) or p62 (bottom) and analysed using confocal microscopy. (B) Intensity profiles were generated for the areas enlarged in (A). (C, D) HeLa cells were treated with 5 μM bortezomib for 4 h, fixed in methanol and stained with a MAR/PAR antibody and p62 (C) or ubiquitin (D) and analysed using confocal microscopy. (E) HeLa (left panel) and U2OS (right panel) cells were incubated with 20 μM MG132, 100 nM bortezomib or 10 μM PR-619 for 3 h, lysed in RIPA lysis buffer supplemented with PARP and PARG inhibitors and analysed using western blot. (F) HeLa (left panel) and U2OS (right panel) cells were treated with 5 μM bortezomib for the indicated durations. Cells were lysed in RIPA buffer supplemented with PARP and PARG inhibitors and analysed using western blot. Source data are available online for this figure.

artefact induced by bafilomycin A treatment, we also performed a knockdown of ATG5 and ATG7, which are essential proteins for formation of the autophagosome. Also, here we see a slightly stronger ADPr signal in both soluble and insoluble fractions (Fig. 3E). Moreover, in this experiment, we used a MAR-specific antibody to confirm the immunofluorescence data showing that MARylation is induced. From these data, we concluded that under specific circumstances, MARylation of proteins can lead to their localisation to p62 bodies, transport to aggresomes and subsequent degradation via autophagy. We did not further analyse processing of ADP-ribosylated proteins to lysosomes, as we and others have noticed before that, depending on amino acid linkage, ADP-ribosylation is highly pH sensitive (Tashiro et al, 2023; Weixler et al, 2023). We would therefore not expect to be able to detect ADP-ribosylated proteins anymore upon fusion of autophagosomes with lysosomes.

## Ubiquitination is necessary for the recruitment of MARylated proteins to p62 bodies

Formation of p62 bodies is highly dependent on ubiquitin, as p62 binds to polyubiquitinated proteins to initiate their sequestration (Bjorkoy et al, 2005). We therefore next asked the question of what happens to ADP-ribosylated proteins in the absence of ubiquitination, as we expect this to interfere with p62 body formation and thus with accumulation and further processing of ADP-ribosylated proteins. We treated cells with TAK-243, an inhibitor of the ubiquitin activating enzyme 1 (UBA1), which prevents both mono- and polyubiquitination. In this condition, we observed no significant ADPr signal and, as expected, no detectable ubiquitinated proteins (Fig. 4A). In contrast to this, using western blot, we observed a substantial accumulation of ADP-ribosylated proteins upon inhibition of ubiquitination (Fig. 4B). A tentative explanation for this phenomenon could be that ADP-ribosylated proteins are recruited to specific foci in a ubiquitin-dependent manner and are highly diffuse without ubiquitin. These diffuse proteins are harder to detect than proteins accumulated in a few foci. In addition to the RIPA soluble proteins, we also analysed the RIPA insoluble protein fraction, as it is possible that the MARylated proteins present in foci are less soluble, similar to, for example, p62 (Danieli and Martens, 2018; Fujita et al, 2011; Johnston and Samant, 2021). In the RIPA insoluble fraction, we found the presence of ADP-ribosylated proteins in diverse cell lines analysed (Fig. 4B). The observed increase in ADP-ribosylated proteins in the western blot implies that PARPs are either activated in response to the stress caused by a lack of ubiquitination or that ubiquitination is required for the further processing of ADP-ribosylated proteins.

To analyse whether any of the PARPs stabilised by proteasome inhibition in an earlier study (Weixler et al, 2023) are influenced by loss of ubiquitination, we generated stable HeLaFlpInTRex with doxycycline inducible expression of mEGFP-tagged PARPs. We incubated these cells with both doxycycline as well as the ubiquitination inhibitor TAK-243. Here we observed a stabilisation of PARP6 and PARP7 and no obvious effects on PARP10, PARP12 and PARP15 (Fig. 4C). Following TAK-243 treatment, we observed an increase in ADP-ribosylation levels in both soluble and insoluble protein fractions. Intriguingly, the only exception is PARP10, where we find less protein ADP-ribosylation in the soluble fraction, but do see an increase in insoluble ADP-ribosylation. PARP10 contains ubiquitin interaction motifs, via which it binds to specific poly-ubiquitin chains (Verheugd et al, 2013; Waltho et al, 2024). It is possible that without polyubiquitin PARP10 cannot localise or bind to its substrates and therefore less proteins are ADP-ribosylated by PARP10 in the absence of ubiquitin. Recent work has shown that the ADPr on PARP10 can be further modified with ubiquitin to form a dual modification termed MARUbylation (Bejan et al, 2025), which is probably the reason why the signal for PARP10 in the TAK-243-treated sample decreases. The diverse PARPs have been suggested to function in various signalling pathways, which in the case of PARP10 might be mediated by its interaction with poly-ubiquitin (Challa et al, 2021; Feijs-Zaja et al, 2024; Hopp and Hottiger, 2021; Luscher et al, 2018). It is thus unlikely that the MARylation introduced by all monoARTs leads to protein degradation.

We hypothesised that promoting protein degradation is a specialised function of one or a few ARTs and therefore performed an experiment where we inhibited several ARTs with diverse cocktails of PARP inhibitors combined with TAK-243 and asked how ADP-ribosylation levels changed. It appears that both PARP7 and TNKS1 are the relevant enzymes in this context, as omission of their inhibitors from the inhibitor cocktail leads to changes in the ADP-ribosylation pattern following TAK-243 incubation (Fig. 4D). The sum of the samples lacking either TNKS1 or PARP7 inhibitors appears equal to the sample without PARP inhibitors, indicating that these two enzymes are the main enzymes responsible for ADP-ribosylation in this condition. This fits with existing data, where PARylation by TNKS1 leads to subsequent polyubiquitination and proteasomal degradation of substrates (Callow et al, 2011; Mariotti et al, 2017; Zhang et al, 2011). By blocking ubiquitination, we interfere with the degradation of TNKS1 substrates and therefore see their accumulation. The question we asked next is whether PARP7, similar to TNKS1, may be involved in the regulation of protein turnover.

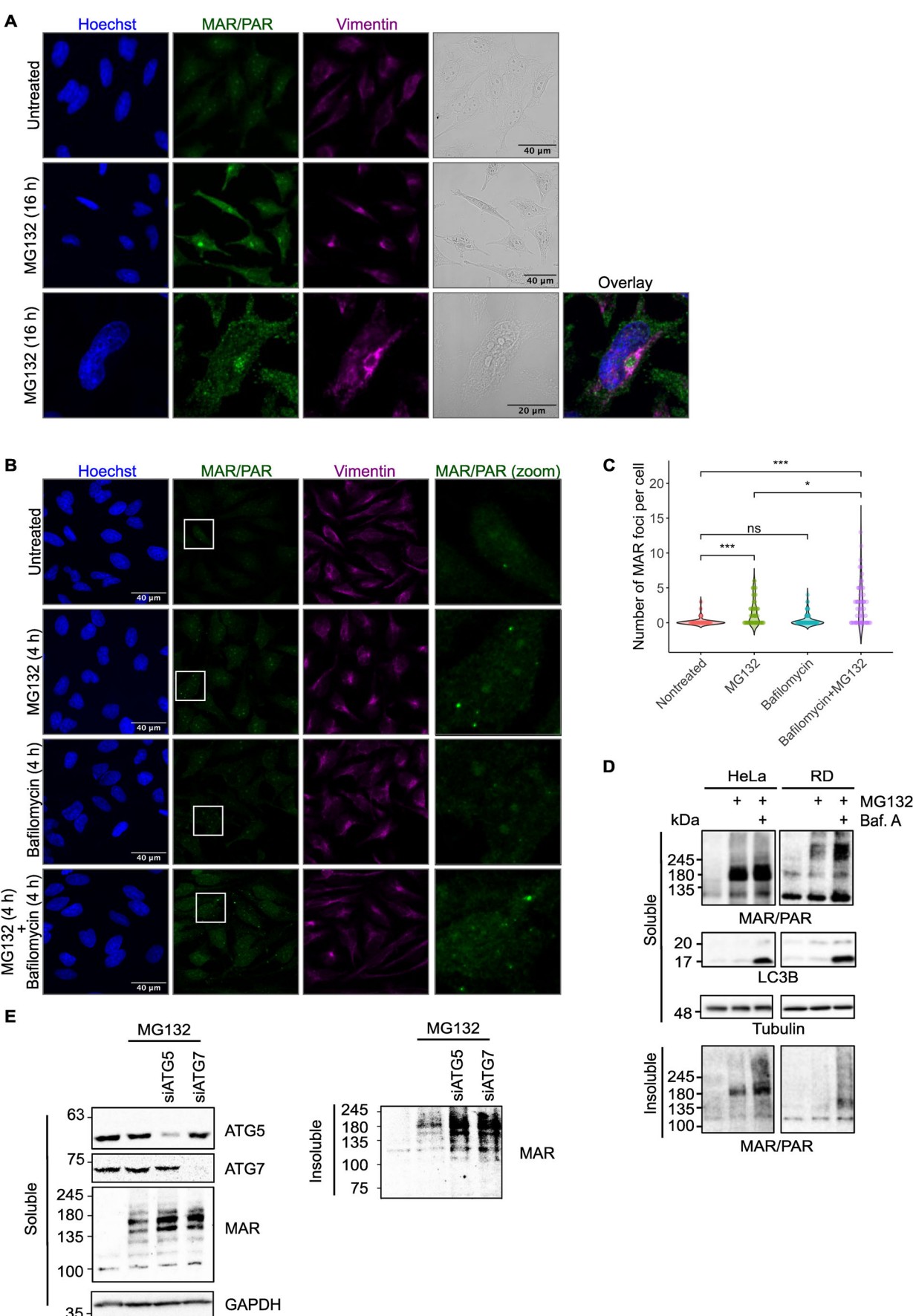

**Figure 3.  MARylated proteins in p62 bodies are transported to aggresomes and degraded via autophagy.**

(A) HeLa cells were treated with 1 μM MG132 for 16 h, fixed and stained using PAR/MAR and vimentin antibodies. The slides were analysed using confocal microscopy. (B) HeLa cells were treated with 1 μM MG132, 10 nM bafilomycin A or both for 4 h, fixed in methanol and stained using PAR/MAR and vimentin antibodies. The slides were analysed using confocal microscopy. (C) Analysis of (B). After masking the nuclear signal, the number of foci per cell was quantified from 30 to 50 cells across 4–5 different images and visualised using a violin plot. Overall differences were assessed using the Kruskal–Wallis test. For post hoc pairwise comparisons, the Mann–Whitney $U$-test was applied and significance levels are indicated as follows: ***$p < 0.001$, *$p < 0.05$ and ns not significant. MG132 vs untreated $p = 0.0004$; MG132 + Bafilomycin vs untreated $p = 0.0001$; MG132 + Bafilomycin vs MG132 $p = 0.027$. (D) HeLa and RD cells were treated with 400 nM MG132 and 12.5 nM bafilomycin A overnight, lysed in RIPA buffer and analysed using western blot with the indicated antibodies. (E) HeLa cells were transfected with siRNA for ATG5 or ATG7 for 48 h, treated with 200 nM MG132 overnight and lysed in RIPA. Samples were analysed using western blot with the indicated antibodies. Both RIPA-soluble and insoluble fractions were analysed. Source data are available online for this figure.

## PARP7 functions as a sensor of proteotoxic stress

When analysing stable, doxycycline-inducible mEGFP-PARP7 HeLa cells, we observed that despite strong induction of other PARPs in this system following addition of doxycycline, mEGFP-PARP7 protein levels are very low to undetectable (Fig. 4C). It has been reported before that inhibition of the proteasome or inhibition of PARP7 activity both lead to an increase in PARP7 protein levels (Lu et al, 2019; Sanderson et al, 2023). When we incubate inducible mEGFP-PARP7 cells with a proteasome inhibitor, PARP7 inhibitor, or a combination of both, we can confirm these results and see an accumulation of mEGFP-PARP7 protein. We also detect PARP7 in the RIPA insoluble fraction in a proteasome inhibition-dependent manner (Fig. 5A). To test whether PARP7 accumulation may be due to off-target effects of the inhibitor, we used a second inhibitor for PARP7, KMR-206, and confirmed that PARP7 stabilisation occurs following inhibition (Appendix Fig. S1). When inhibiting ubiquitination instead of proteasome, we obtain similar results with mEGFP-PARP7 being stabilised by the treatment and present in both soluble as well as insoluble fractions (Fig. 5B). This indicates that PARP7 is degraded by the proteasome in a ubiquitin-dependent manner; however, this is most likely not the only route of PARP7 turnover. It is possible that PARP7 automodification regulates its degradation via alternative mechanisms. Upon co-expression of his-ubiquitin and mEGFP-PARP7 followed by a pull-down of all ubiquitinated proteins, PARP7 is enriched regardless of the presence of a PARP7 inhibitor (Fig. 5C). It thus appears that PARP7 ubiquitination occurs also independent of its catalytic activity, with both ubiquitination and ADP-ribosylation leading to PARP7 protein degradation. One possible explanation for these observations is that the absence of PARP7 is essential for cells in normal conditions and therefore is degraded via a ubiquitin-dependent proteasome degradation pathway, as well as a possible back-up pathway depending on ADP-ribosylation to ensure that the protein is removed from cells unless specific conditions require its activity. Blocking of ubiquitination or proteasomal protein degradation consequently leads to higher PARP7 levels and therefore increased activity. During (patho)physiological conditions leading to dysfunctional protein degradation pathways, PARP7 is rapidly stabilised. The stabilised PARP7 is then able to ADP-ribosylate substrate proteins, which targets those for degradation. To test whether we can observe a stabilisation of proteins when PARP7 is inhibited, we incubated HeLa cells with a PARP7 inhibitor overnight and performed quantitative proteomics. The overall differences between the control and inhibitor-treated cells are small (Fig. 5D; Appendix Fig. S2). Labelled in red are the proteins whose

expression is at least 40% higher in the inhibitor-treated cells, with a corresponding fold change of $\log 2 > 0.5$, implying that these proteins may be regulated by PARP7. Confirming previous experiments, PARP7 itself was the protein by far most affected by the inhibitor treatment ("TIPARP"). To test whether any of the stabilised proteins are indeed modified by PARP7, we performed an enrichment of ADP-ribosylated proteins followed by a western blot with specific antibodies (Fig. 5E). Here, we see a slight enrichment of positive control tubulin (Palavalli Parsons et al, 2021), which was identified as PARP7 substrate previously, as well as c-FOS, which is one of the proteins slightly more stable following PARP7i (Fig. 5D). This experiment indicates that diverse proteins are stabilised following PARP7i, even if under basal conditions the effects on protein stability are small.

If PARP7 is indeed upregulated during proteotoxic stress, the question arises whether PARP7 catalytic activity is required for ADPr foci formation under proteotoxic stress conditions. To test this, we induced foci with a DUB inhibitor and combined this with a generic PARP inhibitor, Phthal01. In this condition, we see a reduction of foci formed, indicating that during stress caused by excessive ubiquitination, ADP-ribosylation activity is essential for p62/ubiquitin condensate formation (Fig. 6A,B). To determine which PARPs are relevant for this, we transfected siRNA against PARP6, PARP7, PARP10 or PARP12, induced foci formation with DUB inhibition and co-stained ADPr and ubiquitin. In this experiment, loss of PARP7 leads to reduced condensate formation, indicating that PARP7 mediates a key function in response to excessive ubiquitination (Fig. 6C,D). In addition, we made use of the PARP7 and TNKS1 inhibitors, RBN2397 and XAV939, respectively, and combined those together with DUBi (Appendix Fig. S3). Confirming the siRNA-based experiments, PARP7i leads to a reduction of foci, whereas TNKSi does not reduce foci, pointing at distinct functions for TNKS1 and PARP7 in this process. Lastly, we asked whether the PARP7 protein, which is transiently present in cells, is sufficient to label proteins for autophagic degradation without an additional stressor. At an incubation of 4 h with bafilomycin A, we did not see a significant increase in cellular ADP-ribosylation (Fig. 3B,C). Therefore, we now incubated cells with a low dose of bafilomycin A overnight. Even though here we have not blocked the proteasome or induced proteotoxic stress in any other way, ADP-ribosylated proteins start to accumulate in foci in this condition (Fig. 6E), presumably due to the inability to be processed further. These proteins represent those that are degraded in an ADP-ribosylation-dependent manner, whose levels are normally quite low and become only visible after prolonged inhibition of autophagy. When we incubate cells with a PARP7 inhibitor in addition to bafilomycin A, the ADPr-

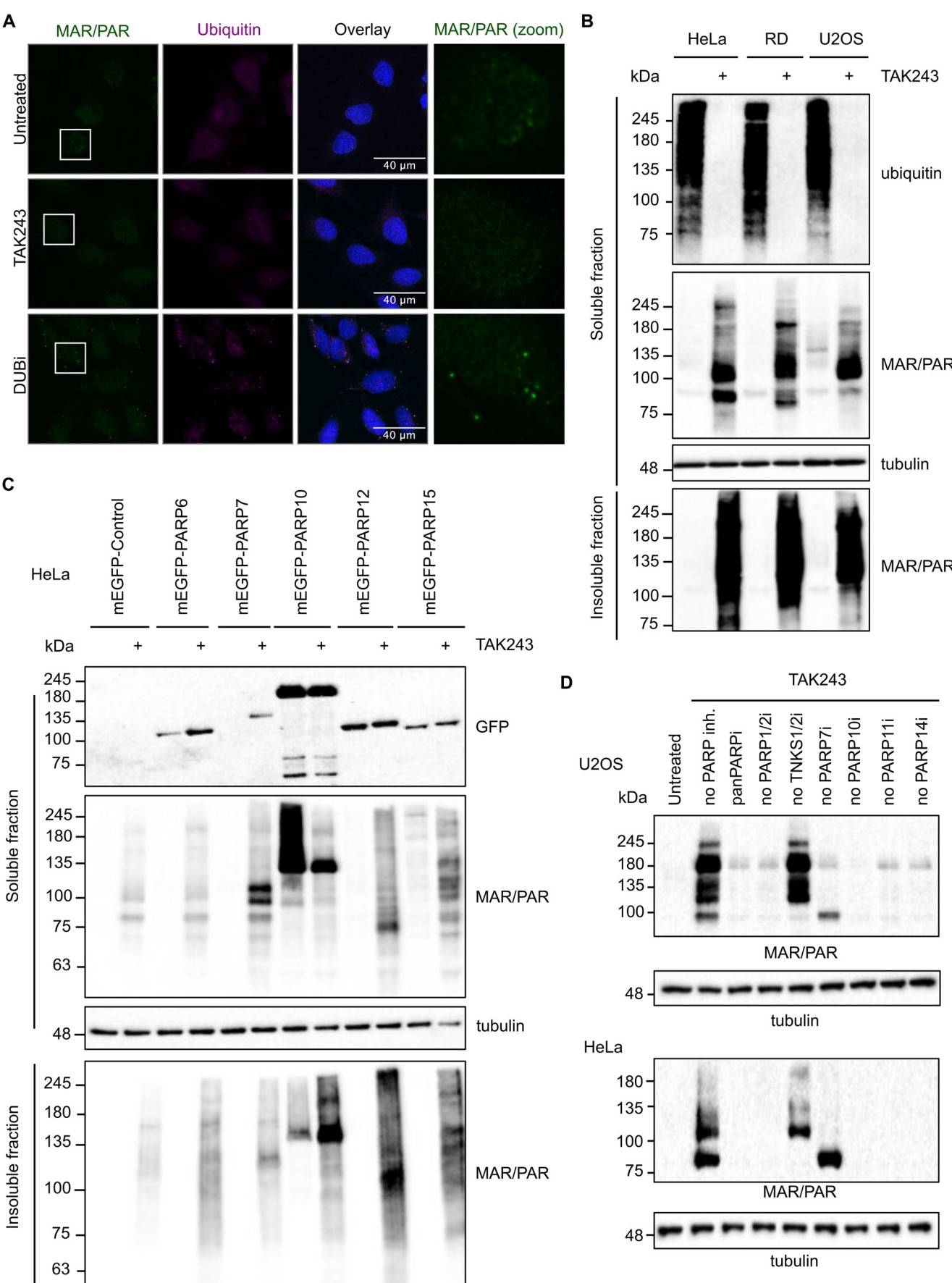

**Figure 4.   Loss of ubiquitination stabilises several PARPs and leads to increased levels of insoluble MARylated proteins.**

(**A**) HeLa cells were treated with 2 µM TAK-243 for 5 h, fixed and stained using MAR/PAR and ubiquitin antibodies. The samples were analysed using confocal microscopy; the scale bar represents 40 µm. (**B**) HeLa, RD and U2OS cells were incubated with 1 µM TAK-243 for 4 h, followed by lysis in RIPA buffer. The insoluble pellet was resuspended in Laemmli sample buffer. Both fractions were analysed using western blot and ubiquitin, MAR/PAR and tubulin antibodies. From the total cell lysate 13% was loaded on the gel, and from the insoluble fraction 40% was loaded. (**C**) HeLaFlpInTRex cells inducibly expressing mEGFP, mEGFP-PARP6, mEGFP-PARP7, mEGFP-PARP10, mEGFP-PARP12 or mEGFP-PARP15 were seeded in 6-well plates in the presence of doxycycline to induce mEGFP-construct expression. 24 h after doxycycline addition, cells were incubated with 1 µM TAK-243 for 4 h, lysed in RIPA lysis buffer and analysed using western blot with GFP, MAR/PAR and tubulin antibodies. From the total cell lysate 13% was loaded on the gel, and from the insoluble fraction 40% was loaded. (**D**) HeLa and U2OS cells were treated overnight with 100 nM TAK-243 combined with the indicated PARP inhibitor cocktail termed "panPARPi" (each at a concentration of 75 nM except for PARP10 inhibitor, which is at a concentration of 750 nM), or a cocktail omitting the indicated PARP inhibitors. Cells were lysed in RIPA buffer and analysed using western blot with a MAR/PAR and tubulin antibody. PARP inhibitors used: Olaparib (PARP1i/PARP2i), XAV939 (TNKS1i/TNKS2i/PARP2i), RBN2397 (PARP7i), OUL35 (PARP10i), ITK7 (PARP11i), RBN12759 (PARP14i). Source data are available online for this figure.

containing foci do not form, confirming that the observed accumulation of ADP-ribosylated proteins is dependent on PARP7 (Fig. 6E). The absence of PARP7 activity did not interfere with LC3-containing foci in general, implying that ADP-ribosylation as such is not essential for functional autophagy and thus has a clearly distinct function from other PTMs in this process, as for example ubiquitin is absolutely required.

## Discussion

Both PARP7 and TNKS1 have been described to modify specific proteins, which subsequently get degraded (Feijs-Zaja et al, 2024). Under conditions where protein homoeostasis is perturbed, such as a dysfunctional proteasome or interference with ubiquitination, the PARP7 protein stabilises and rapidly accumulates (Fig. 5). Proteins MARylated by PARP7 are subsequently degraded via an autophagosomal pathway, as indicated by their accumulation following blocking of autophagy (Fig. 6E). Once the initial stress abates and protein homoeostasis is balanced again, PARP7 itself will also be degraded to not interfere with normal cellular processes. This hypothesis is further supported by the fact that following proteotoxic stress, ubiquitination accumulates immediately whereas the signal for ADP-ribosylation increases with a delay (Fig. 2F). Ubiquitinated proteins that are destined for proteasomal degradation appear immediately following inhibition of the proteasome, whereas the MARylated proteins are generated in response to this proteotoxic stress and are thus detectable at a later time point only. Although our work suggests a role for MARylation in protein degradation pathways, many questions still need to be addressed, as we, for example, do not yet know how exactly ADPr and ubiquitin co-regulate this process, nor do we know whether MARylation could also target proteins for proteasomal degradation. It is clear from our work that the level of MARylated proteins increases in response to proteotoxic stress as well as following inhibition of either ubiquitination or autophagy, and that PARP7 plays a major role in this process.

PARP7 was originally identified as a gene inducible by 2,3,7,8-tetrachlorodibenzodioxin (TCDD, henceforth referred to as "dioxin") and is therefore also known as TiPARP for "TCDD inducible poly(ADP-ribose)polymerase" (Ma et al, 2001). Upon dioxin stimulation, PARP7 MARylates AHR, leading to its degradation (MacPherson et al, 2013; Rijo et al, 2021). PARP7 RNA expression is furthermore induced by HIF1a and oestrogen

(Rasmussen et al, 2021; Zhang et al, 2020). Several works have also shown that inhibition of the proteasome leads to a significant increase in PARP7 protein levels, indicating that under normal conditions its proteasomal turnover is high (Lu et al, 2019; Weixler et al, 2023). Following induction by HIF1a, PARP7 forms condensates in the nucleus to which HIF1a is recruited and subsequently polyubiquitinated (Zhang et al, 2020). Both dioxin exposure and hypoxia can lead to the formation of reactive oxygen species, with the consequence of protein damage. Stabilisation of PARP7 in this condition may be required for clearance of the damaged proteins that arise (Fig. 7A,B). Once the normal protein homoeostasis has been restored, PARP7 will be rapidly degraded to prevent unwanted ADP-ribosylation in healthy cells.

Upon inhibition of proteasomal degradation, at early time points, we observe an accumulation of ADP-ribosylated proteins in foci positive for p62 and ubiquitin, which we presume to be p62 bodies, also known as ALIS bodies. In the absence of ubiquitination, we observe an accumulation of ADP-ribosylated proteins in cell lysates, with a significant proportion of the ADP-ribosylated proteins present in the RIPA-insoluble fraction (Fig. 4B). It is possible that ADPr serves as a mark to label superfluous, misfolded or damaged proteins, which are then subsequently poly-ubiquitinated and degraded. Without subsequent ubiquitination, MARylated proteins are not further processed and therefore accumulate (Fig. 7C). It is thinkable that similar to TNKS1 and PARylation, also for MARylation specific recognition modules exist within ubiquitin E3 ligases to mediate ubiquitination of MARylated substrates. Not only PARylation, but also MARylation could thus form a signal indirectly leading to the degradation of modified proteins. It is well possible that specific ubiquitin E3 ligases are recruited to MARylated proteins via ADPr binding domains, providing a tentative mechanism for E3 ligase substrate recognition. It has been shown that the macrodomains of PARP9 can recruit it to proteins modified by PARP7, which will in turn bring in its interaction partner ubiquitin E3 ligase deltex 3 (DTX3L) (Juszczynski et al, 2006; Yang et al, 2021). Subsequent poly-ubiquitination of the MARylated proteins can then occur. Considering the increase in ADP-ribosylation following prolonged inhibition of autophagy, and the absence thereof when PARP7 is inhibited (Fig. 6E), we suggest that MARylation by PARP7, possibly followed by ubiquitination, forms a generic protein regulatory mechanism. Under basal conditions, this happens at low levels and becomes visible only after prolonged inhibition of autophagic turnover; however, when additional

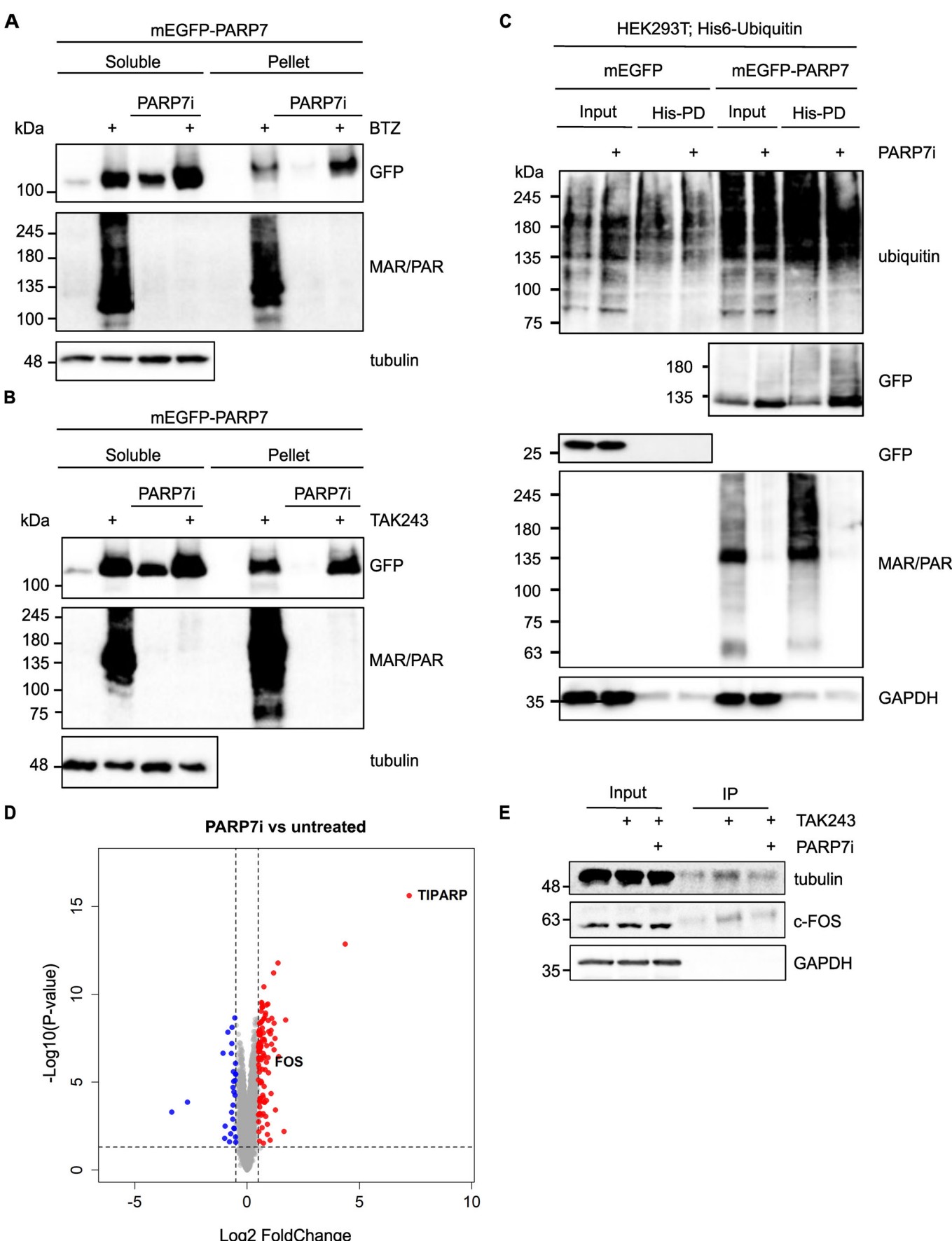

**Figure 5.   PARP7 is stabilised by proteotoxic stress.**

(A) HeLaFlpInTRex-mEGFP-PARP7 cells were incubated overnight with 100 nM XAV939 (all samples), 400 ng/ml doxycycline (all samples), 200 nM bortezomib and 200 nM RBN2397 (PARP7i) as indicated. Cells were lysed in RIPA lysis buffer, and the insoluble pellet was resuspended in Laemmli sample buffer. Samples were analysed using western blot with GFP, PAR/MAR and tubulin antibodies. From the total cell lysate 13% was loaded on the gel, and from the insoluble fraction 40% was loaded. (B) HeLaFlpInTRex-mEGFP-PARP7 cells were incubated overnight with 100 nM XAV939 (all samples), 400 ng/ml doxycycline (all samples), 50 nM TAK243 and 200 nM RBN2397. Cells were lysed in RIPA lysis buffer, and the insoluble pellet was resuspended in Laemmli sample buffer. Samples were analysed using western blot with GFP, PAR/MAR and tubulin antibodies. From the total cell lysate 13% was loaded on the gel, and from the insoluble fraction 40% was loaded. (C) HEK293T cells were transfected with His-tagged ubiquitin and mEGFP-PARP7. 24 h after transfection, cells were incubated with 100 nM PARP7 inhibitor RBN2397 overnight, followed by a 3-h incubation with 5 μM MG132. Cells were lysed in 8 M urea, followed by a His tag pull-down of ubiquitinated proteins. Samples were analysed using western blot with ubiquitin, GFP and PAR/MAR antibodies. (D) HeLa cells were treated with 100 nM RBN2397 overnight, followed by quantitative mass-spectrometric analysis of 4 replicates per condition. A Volcano plot was generated to visualise the resulting changes in protein abundance. Statistical significance was assessed using the empirical Bayes-moderated two-sided *t* test implemented in the limma package (R). The *y*-axis shows the negative $\log_{10}$ of the *p* values. (E) HeLa cells were treated with 100 nM RBN2397 overnight, followed by lysis and enrichment of ADP-ribosylated proteins using a PAR/MAR antibody. Immunoprecipitated proteins were analysed using western blot, where specific proteins were detected as indicated. The contrast on the tubulin blot was increased to better visualise the signal in the IP samples. Source data are available online for this figure.

stress is induced by blocking the proteasome or interfering with the ubiquitination system, this modification becomes more pronounced. In the presence of excessive ubiquitination induced by inhibition of deubiquitinating enzymes, ADP-ribosylation by PARP7 appears required for the formation of foci containing ubiquitin and p62. It is not clear yet why ADP-ribosylation becomes essential during DUBi, as we could not measure an effect on autophagic flow in general following PARP inhibition. It is possible that p62 itself forms a relevant ADP-ribosylation substrate in this context, as it was reported to be modified by PARP14 (Kubon et al, 2024). In our MS dataset p62 stability was only slightly different by PARP7 inhibition (±1.2 fold down-regulated), nevertheless p62 could form a direct PARP7 substrate following stress.

A possible function of this ADP-ribosylation could be to assist with the segregation of unwanted proteins, possibly by lowering their solubility. It has been described that proteins destined for proteasomal degradation tend to form protein condensates or insoluble protein inclusion bodies when they cannot be degraded. These condensates are dynamic structures that can dissolve upon stress relief, whereas inclusions are irreversible and serve to separate the potentially toxic proteins from the rest of the cell (Enenkel et al, 2022). It will be intriguing to determine in future work whether ADP-ribosylation is a key factor required to protect cells from toxic proteins by enabling their segregation. We did observe an increase in ADP-ribosylated proteins following inhibition of ubiquitin in the RIPA-insoluble fraction (Fig. 4B), but failed to localise the signal using immunofluorescence microscopy approaches. It is thus not clear yet whether the proteins MARylated by PARP7 are present in aggregates or may be dispersed and therefore not readily detectable. Recent work has identified ADPr-containing foci which colocalise with p62 and ubiquitin that form in a PARP14-dependent manner following interferon stimulation (Kar et al, 2024; Kubon et al, 2024; Ribeiro et al, 2024). Despite the efforts of several labs, the function of these foci is hitherto unclear. The PARP14-dependent foci appear to be clearly distinct from the foci we identified, as here no co-staining with LC3B was visible (Raja et al, 2025). Inhibition of the proteasome had an opposite effect on PARP14-dependent foci, as their formation was impeded following interference with protea-somal degradation (Raja et al, 2025). Intriguingly, it was recently shown that in cells (Bejan et al, 2025), ubiquitin can be transferred

directly onto ADP-ribose present on proteins to form a dual modification termed MARUbylation, which was previously shown to occur in vitro (Zhu et al, 2022). Two recent preprints indicate that PARP7-modified proteins are MARUbylated by DTX2 and that this modification can be detected by specific readers such as the ubiquitin E3-ligase RNF114 (Kloet et al, 2025; Lacoursiere et al, 2025). This E3-ligase labels proteins with K11-linked polyubiquitin, which has been ascribed a role in protein degradation. It is thus possible that the ADP-ribosylation introduced by PARP7 forms a platform for further modification with ubiquitin, which will ultimately lead to degradation of modified proteins.

One of the questions that we have asked before (Feijs and Zaja, 2022) is how a handful of monoARTs can potentially modify thousands of proteins, as identified by various MS-based studies (Buch-Larsen et al, 2021; Larsen et al, 2018; Martello et al, 2016), and still be considered specific. We believe it is possible that during specific cellular stress, the induced MARylation is not specific but instead may simply occur on exposed, accessible amino acids of proximal proteins, enabling the further processing or segregation of potentially harmful misfolded proteins. Our work strongly suggests that MARylation is involved in protein turnover in a ubiquitin-dependent manner. Another outstanding question in the field regards the identity of the eraser of cysteine-MARylation introduced by PARP7, which has not been identified to date despite efforts from several labs (Sowa et al, 2021; Weixler et al, 2025). One of the possibilities raised by the current work is that perhaps such an eraser is not required. If PARP7 labels substrates for degradation via autophagy, ultimately MARylated proteins end up in a highly acidic environment upon fusion with lysosomes, where the modification may be lost non-enzymatically, which provides a tentative explanation for the apparent lack of a cysteine-specific ADP-ribosylhydrolase.

## Limitations

We have used quite harsh cell treatments to investigate the role of PARP7 during proteotoxic stress, including generic inhibition of ubiquitination or deubiquitination systems. More work is needed to decipher which (patho)physiological stressors lead to PARP7 activation, and in those conditions, follow the degradation of PARP7 substrates to determine the exact pathway followed. Furthermore, the PARP7 protein analysed here was overexpressed

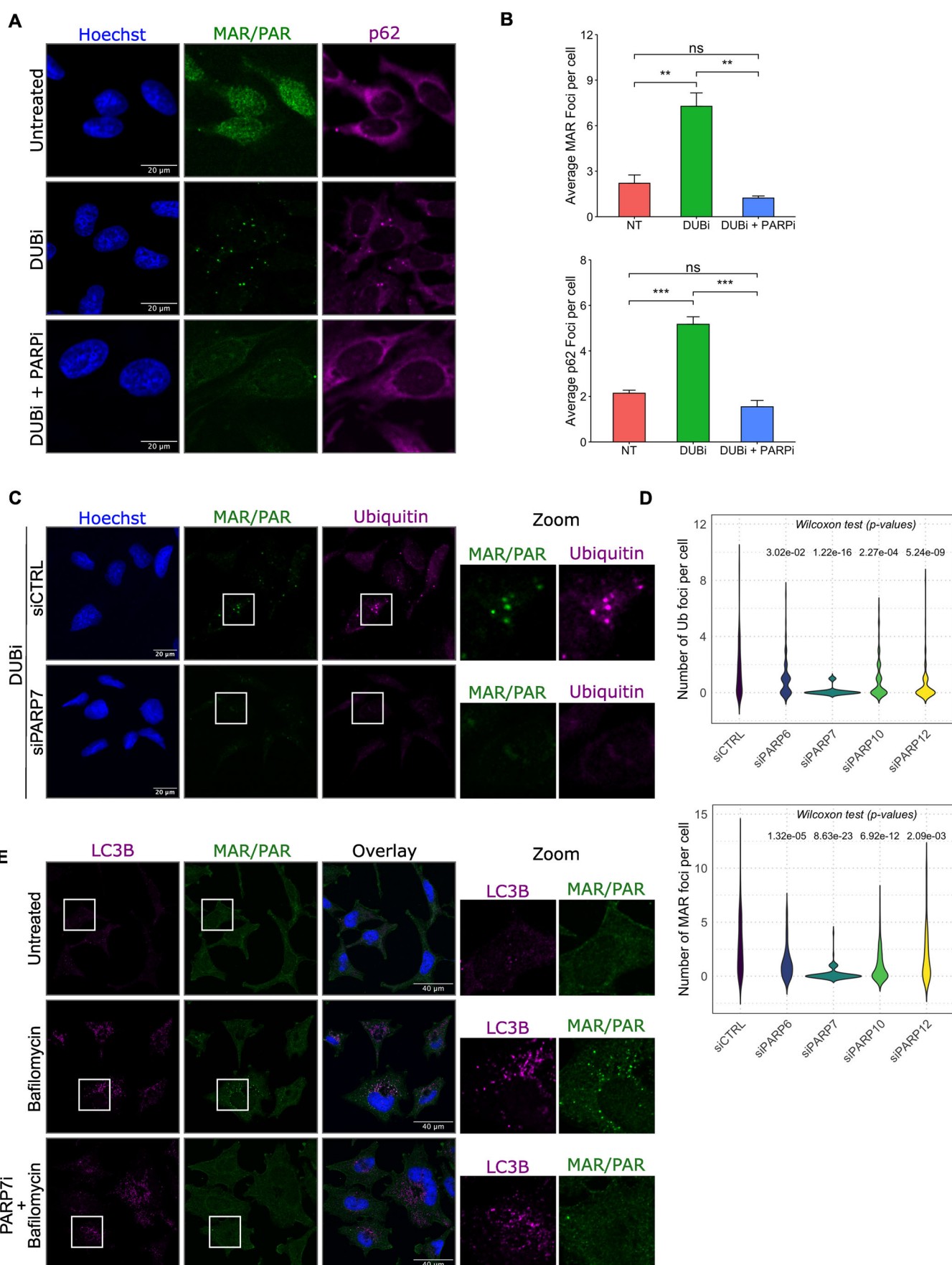

**Figure 6.   Inhibition or loss of PARP7 prevents accumulation of ADP-ribosylated proteins during proteotoxic stress.**

(**A**) HeLa cells were seeded on glass coverslips and treated with 10 µM DUB inhibitor PR-619 and 10 µM generic PARP inhibitor Phthal01 for 30 min as indicated. Following treatment, cells were fixed using glyoxal and stained with p62 and PAR/MAR antibodies. As observed in Fig. 1, glyoxal fixation leads to higher nuclear staining. (**B**) Quantification of the foci formed in the cells in (**A**). The number of foci per image frame containing 10–20 cells was counted across 4–5 different images. After counting the nuclei per frame, the number of foci was divided by the number of nuclei per frame and the average number of foci per nuclei/cell was plotted. Differences were first assessed using the Kruskal–Wallis test. For post hoc pairwise comparisons, the Mann–Whitney *U*-test was applied, and significance levels are indicated as follows: \*\*\**p* < 0.001, \*\**p* < 0.01, and ns not significant. MAR foci: DUBi vs NT *p* = 0.0068; DUBi + PARPi vs DUBi *p* = 0.0062. p62 foci: DUBi vs NT *p* = 0.0009; DUBi + PARPi vs DUBi *p* = 0.0002. Error bars represent standard deviations. (**C**) HeLa cells were seeded on glass coverslips, transfected with the indicated siRNAs and treated with 10 µM PR-619. Cells were fixed using methanol and stained with an ubiquitin and PAR/MAR antibody, followed by image acquisition using a confocal microscope. (**D**) Quantification of the foci formed in the cells in (**C**). After masking the nuclear signal, the number of foci per cell was quantified from 30 to 50 cells across 4–5 different images and visualised using a violin plot. Overall differences were assessed using the Kruskal–Wallis test and p-value displayed on the plot, with the number of ubiquitin foci (top) and ADPr foci (bottom) analysed. (**E**) HeLa cells were treated with 10 nM bafilomycin A overnight in the presence or absence of 100 nM PARP7 inhibitor RBN2397. Cells were fixed and stained using LC3B and ADPr antibodies, followed by analysis using a confocal microscope.

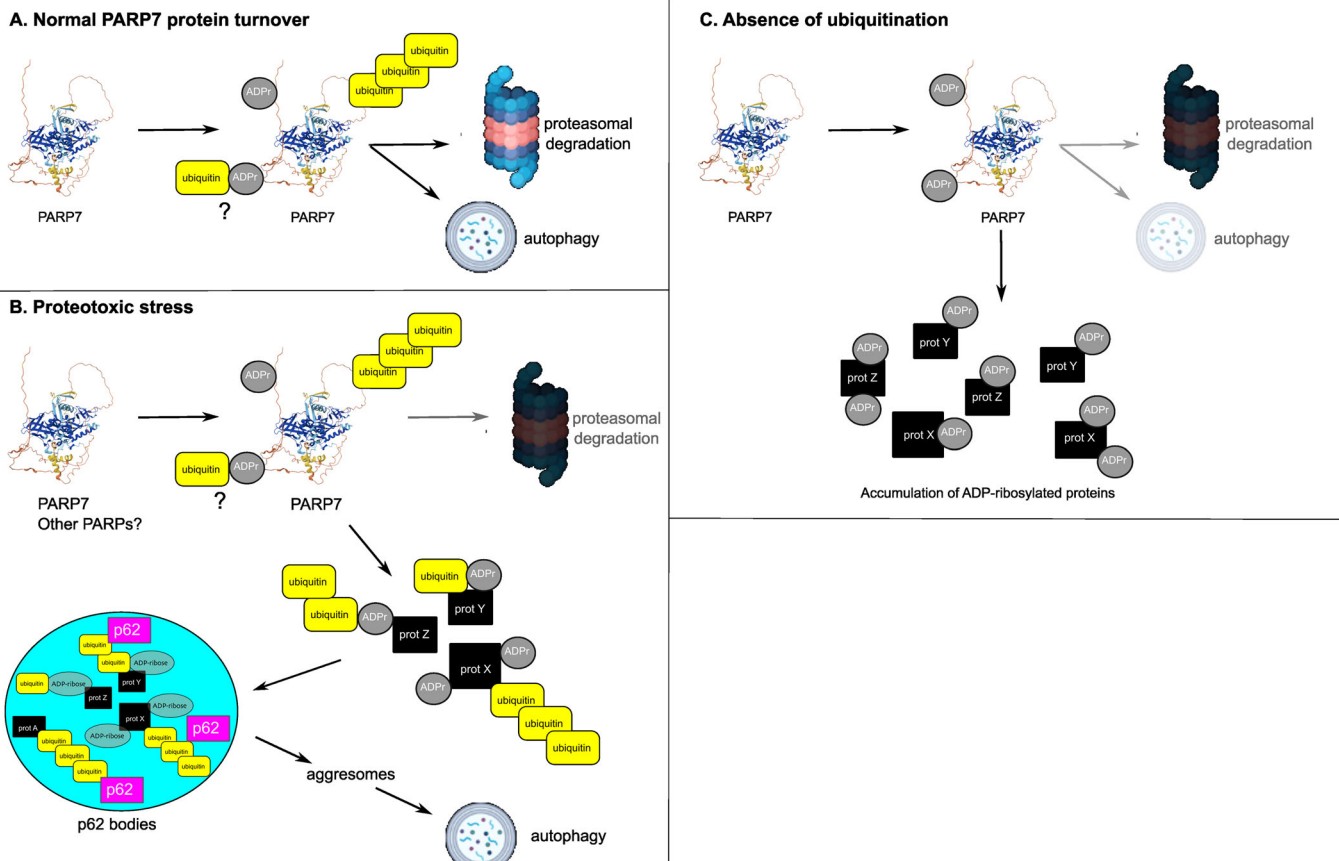

**Figure 7.   Schematic overview of the proposed role of ADP-ribosylation and PARP7 in regulation of protein homoeostasis.**

(**A**) In normal conditions, PARP7 is rapidly degraded via both proteasome and autophagy, regulated by both ADP-ribosylation as well as ubiquitination. The question mark indicates that it may be possible that ubiquitin is also directly attached to ADPr, but this has not been tested in this study. (**B**) During circumstances where proteasomal function is impaired, the PARP7 protein is stabilised. PARP7 modifies substrate proteins, which are then degraded via autophagy in a ubiquitin-dependent manner. It is possible that in addition to PARP7, other PARPs have similar functions. (**C**) In the absence of ubiquitination, PARP7 cannot be degraded via neither the proteasome nor autophagy, leading to strong accumulation of the protein. The proteins that are modified by PARP7 can likewise not be degraded and accumulate in the cells. Considering their presence in insoluble fractions during RIPA lysis, it is possible that ADP-ribosylated proteins segregate in insoluble structures. This hypothesis needs to be further tested, as we were unable to observe such structures using immunofluorescence.

and mEGFP-tagged. Ideally, key findings should be confirmed with antibodies that detect the endogenous protein. We were unable to do this at this moment, as none of the antibodies we tested recognised PARP7.

# Methods

### Reagents and tools table

| Reagent/ resource | Reference or source | Identifier or catalogue number | |
|---|---|---|---|
| Experimental models | | | |
| HeLa | Bernhard Lüscher | ATCC CCL-2 | |
| HeLa-Flp-In T-REx | Stephen S. Taylor | | |
| HEK293T | Bernhard Lüscher | ATCC CRL-3216 | |
| U2OS | Bernhard Lüscher | ATCC HTB-96 | |
| RD | Karl Morten | ATCC CCL-136 | |
| Recombinant DNA | | | |
| pcDNA5-FRT/TO-N-mEGFP-PARP6 | Addgene | 178005 | |
| pcDNA5-FRT/TO-N-mEGFP-PARP7 | Addgene | 178004 | |
| pcDNA5-FRT/TO-N-mEGFP-PARP10 | Addgene | 177997 | |
| pcDNA5-FRT/TO-N-mEGFP-PARP12 | Addgene | 177999 | |
| pcDNA5-FRT/TO-N-mEGFP-PARP15 | Addgene | 178000 | |
| pOG44 | Invitrogen | V600520 | |
| Antibodies | | | Dilution |
| Anti-ADPr HCA354 | BioRad | abD33204 | WB 1:1000, IF 1:500 |
| Anti-MAR/PAR (E6F6A) | Cell Signaling Technology | 83732S | WB 1:100,000, IF 1:5000 |
| Anti-ubiquitin (P4D1) | Santa Cruz | sc-8017 | WB 1:1000, IF 1:250 |
| Anti-mono and polyubiquitinated conjugates (FK2) | BIOZOL | UBI-68-0122-100 | WB 1:2000, IF 1:1000 |
| Anti-tubulin-alpha | Santa Cruz | sc-5286 | WB 1:1000 |
| Anti-p62 | Santa Cruz | sc-28359 | WB 1:1000, IF 1:250 |
| Anti-GFP | Chromotek | pabg1 | WB 1:2000 |
| Anti-HA | Covance | MMS-101R | WB 1:1000 |
| Anti-LC3B | Santa Cruz | sc-376404 | IF 1:1000 |
| Anti-LC3B | Cell Signaling Technology | 2775S | WB 1:1000 |
| Anti-GAPDH | Santa Cruz | sc-32233 | WB 1:2000 |
| Anti-c-FOS | Proteintech | 66590-1-Ig | WB 1:1000 |
| Anti-vimentin | Sigma | V6389 | IF 1:250 |

| Reagent/ resource | Reference or source | Identifier or catalogue number | |
|---|---|---|---|
| Anti-ATG5 | Cell Signaling Technology | 12994 | WB 1:1000 |
| Anti-ATG7 | Cell Signaling Technology | 8558 | WB 1:1000 |
| Oligonucleotides and other sequence-based reagents | | | |
| siPARP6 | Dharmacon | M-013797-02-0005 | |
| siPARP7 | Dharmacon | M-013948-03-0005 | |
| siPARP10 | Dharmacon | M-014997-03-0010 | |
| siPARP12 | Dharmacon | M-013740-01-0010 | |
| siPARP15 | Dharmacon | M-017186-00-0005 | |
| siATG5 | Dharmacon | M-004374-04-0005 | |
| siATG7 | Dharmacon | M-020112-01-0005 | |
| siControl | Dharmacon | D-001206-14-20 | |
| Chemicals, enzymes, and other reagents | | | |
| MG132 | Calbiochem | CAS 133407-82-6 | |
| TAK-243 (MLN7243) | Biomol | CAY30108-1 | |
| PR-619 | Biomol | CAY16276-5 | |
| Bortezomib | Biomol | CAY10008822-5 | |
| Bafilomycin A | Biomol | Cay11038-500 | |
| Phthal01 | Michael Cohen | | |
| KMR-206 | Michael Cohen | | |
| RBN2397 | MedChem Express | HY-136174 | |
| RBN012759 | MedChem Express | HY-136979 | |
| OUL35 | Tocris | 6344 | |
| ITK7 | Sigma-Aldrich | SML2669 | |
| Olaparib | Selleckchem | S1060 | |
| XAV939 | Selleckchem | S1180 | |
| Prolong Diamond Anti-Fade | ThermoFischer | P36961 | |
| Software | | | |
| Fiji | https://imagej.net/software/fiji/ | | |
| CellProfiler | https://cellprofiler.org | | |
| R studio version 2024.4.2.764 | https://cran.r-project.org | | |
| Inkscape | https://inkscape.en.softonic.com/mac/download | | |
| Other | | | |

| Reagent/ resource | Reference or source | Identifier or catalogue number | |
|---|---|---|---|
| LSM710 Confocal Laser Scanning Microscope equipped with an AxioCam (Zeiss) and a C-Apochromat ×20 objective | Zeiss | | |

## Plasmids and antibodies

Generation of pcDNA-based mEGFP-tagged PARP plasmids was described before (Weixler et al, 2023) and is available from Addgene, the His-tagged ubiquitin construct was described earlier (Verheugd et al, 2013). Antibodies used: rabbit anti-ADPr (HCA354, BioRad), rabbit anti-MAR/PAR E6F6A (CST, lot 5; Cat 83732S), mouse anti-ubiquitin P4D1 (Santa Cruz, sc-8017, Lot D0123), mouse anti-mono and polyubiquitinated conjugates FK2 (BIOZOL, Cat UBI-68-0122-100), mouse anti-Tubulin-alpha (Santa Cruz, sc-5286, Lot L1522), mouse anti-p62 (Santa Cruz, sc-28359, Lot H2621), rabbit anti-GFP (Chromotek, Cat pabg1, Lot 70828032AB), mouse anti-HA (Covance, Cat MMS-101R), mouse anti-LC3B (G-9) (Santa Cruz, Cat sc-376404, Lot G2523), Rabbit anti-LC3B (CST, Cat 2775S, Lot 14), mouse anti-GAPDH (Santa Cruz, Cat sc-32233, Lot J2523) and rabbit anti-c-FOS (66590-1-Ig, Proteintech). All antibodies were used at 1:1000 for western blot and 1:500 for immunofluorescence unless stated otherwise.

## Mammalian cell culture and transfection

HeLa Flp-In T-Rex cells were a gift from Steven Taylor (University of Manchester, Manchester, UK). HEK293T (ATCC CRL-3216), U2OS (ATCC HTB-96) and HeLa cells were cultured in DMEM with pyruvate, 4.5 g/l glucose (Gibco), and 10% heat-inactivated foetal calf serum (Gibco) in a humidified atmosphere with 5% $CO_2$. Cells were frequently tested for mycoplasma contamination and confirmed negative at the moment of the experiments. The calcium phosphate $(Ca_3(PO_4)_2)$ transfection protocol was used to transfect plasmid DNA into mammalian cells. A detailed, step-by-step transfection protocol is available online (Feijs, 2021a). Cells were washed 24 h after transfection in warm HEPES, followed by lysis or fixation of the cells 48 h after transfection. siRNA was transfected using Lipofectamine RNAiMAX as per the manufacturer's instructions at a final siRNA concentration of 10 nM. Cells were washed in PBS 6 h post-transfection and incubated in full cell culture medium for an additional 72 h before the analysis.

## Generation of stable cell lines

HeLa Flp-In T-REx cells were transfected with pcDNA5/FRT/TO-N-mEGFP, -PARP6 (Addgene 178005), -PARP7 (Addgene 178004), -PARP10 (Addgene 177997), -PARP12 (Addgene 177999), PARP15 (Addgene 178000) and pOG44 (Invitrogen). Transfections were performed with calcium phosphate as described in a detailed protocol (Feijs, 2021a). The transfected cells were selected using 5 µg/ml blasticidin and 200 µg/ml hygromycin, kept under selection during further culturing (5 µg/ml blasticidin and 100 µg/ml hygromycin) and were kept as polyclonal cell lines. The inducible expression of mEGFP-proteins was confirmed using western blotting. All cells used were regularly tested for mycoplasma contamination and only used when confirmed mycoplasma-free.

## His-ubiquitin pull-down

HEK293T cells were seeded in 10 cm culture dishes and transfected using calcium phosphate, followed by washing with warm HEPES the next day. Forty-eight hours after transfection, cells were treated with diverse compounds as indicated and lysed in 1 ml urea buffer (8 M Urea, 0.3 M NaCl, 0.1 M sodium phosphate pH 8.0, 10 mM Imidazole) at room temperature. The lysate was sonicated twice with 30 pulses at 60% amplitude. This was followed by centrifugation at maximum rpm for 15 min at room temperature. Soluble fractions were collected for His-ubiquitin pull-downs. Cobalt beads were equilibrated with urea buffer and added to the lysates (20 µl bead slurry per lysate). This was followed by incubation for 2-3 h at room temperature on an overhead rotor. The beads were washed 4× with urea buffer (2 min; 700 g). Finally, the beads were resuspended in 1× sample buffer containing 500 nM imidazole. The beads were heated at 75 °C for 5 min, and the samples were probed by Western blotting.

## Western blotting

Cell protein extractions were performed using urea buffer as described above or using RIPA buffer (150 mM NaCl, 1% Triton X-100, 0.5% sodium deoxycholate, 0.1% SDS, 50 mM Tris-HCl (pH 8.0)) supplemented with protease inhibitor cocktail (Sigma), olaparib (Selleck Chemicals) and benzonase (Sigma). Lysates were centrifuged at $13,000 \times g$ and supernatants collected. The resulting pellets were washed twice in RIPA and resuspended in Laemmli buffer to be able to analyse the RIPA-insoluble fractions. Soluble and insoluble fractions were analysed on home-made tris-glycine SDS-polyacrylamide gels and subsequently blotted onto nitrocellulose membranes using a BioRad TurboBlot system. Membranes were blocked with 5% non-fat milk in TBST for 1 h at RT, primary antibodies were diluted in TBST and incubated overnight at 4 °C, secondary antibodies were diluted 1:10,000 in 5% non-fat milk in TBST and incubated for 1 hr at RT. Wash steps were performed in between and after antibody incubations with TBST at RT for at least 5 min. Blots were detected using the Azure600 detection system.

## Immunofluorescence microscopy

Cells were seeded onto glass coverslips in 24-well plates and treated with diverse compounds as indicated in the figure legends. Following cell treatments, cells were washed once with warm DMEM without FCS and fixed for 20 min with 4% paraformaldehyde in PBS at room temperature. Alternatively, ice-cold methanol was added to the cells on coverslips after removal of the growth medium, followed by 5 min incubation on ice and subsequent quick washing in PBS. For glyoxal fixation, glyoxal solution was added to the cells for 30 min on ice followed by 30 min at room temperature.

After washing in PBS, the samples were quenched using 100 mM ammonium chloride and finally permeabilization using 0.1% Triton X-100. Regardless of fixation method, subsequent blocking was done in PBS supplemented with 1% BSA in PBS with 0.1% Triton X-100 for 1 h at RT. Primary and secondary antibodies were applied for 1 h at room temperature, with extensive washing in between. Secondary antibodies used: AlexaFluor488 anti-mouse, AlexaFluor488 anti-rabbit, AlexaFluor594 anti-mouse and Alexa-Fluor594 anti-rabbit (ThermoFisher) used at 1:2000 in PBS with 0.2% BSA. Following extensive washing in PBS and demineralised water, cells were incubated in a Hoechst-containing solution, followed by mounting using Prolong Anti-Fade Diamond Mountant (ThermoFisher). The samples were analysed with a Zeiss LSM710 Confocal Laser Scanning Microscope equipped with an AxioCam (Zeiss) and a C-Apochromat ×20 objective. Step-by-step immunofluorescence protocols are available online (Feijs, 2021b).

## Mass spectrometry

Frozen cell pellets (untreated vs PARP7i) were lysed in lysis buffer provided by the EasyPep kit (A40006, Thermo Scientific) with ultrasonic disruption (Bioruptor® Pico sonication device). Protein concentration was determined using the Pierce BCA Protein Assay Kit (23227, Thermo Scientific). In all, 100 μg of protein lysate of each sample was proteolytically digested using the EasyPep protocol as provided by the manufacturer. The peptides were dried down and stored at −80 °C. Prior to MS analysis, the lyophilised peptides were resuspended in 3% formic acid (FA)/1% acetonitrile (ACN). The samples were first loaded onto a trapping column (174500, PepMap Neo, C18, 5 μm, 300 μm i.d. × 5 mm, Thermo Scientific) for 10 min. This was followed by peptide separation on an analytical column (Aurora Ultimate 25*75 C18 UHPL, AUR3-25075C18, IonOpticks) at 45 °C employing a 160 min gradient: 0–2 min: 1–2% buffer B (buffer A: 0.1% FA; buffer B: 80% ACN, 0.1% FA), 10–105 min: 30% buffer B, 105–130 min: 30–40% buffer B, 130–137 min: 40–99% buffer B, 137–142 min: 99% buffer B, 142–145 min: 99–2% buffer B, 145–160 min: 2% buffer B at 250 nl/min.

Data acquisition was performed on an Exploris 480 mass spectrometer (Thermo Scientific) using a data-independent acquisition (DIA) approach. A staggered windows mode was used (isolation window of 8 $m/z$, shift of 4 m/z; MS1 scans were carried out from 390 to 1010 or from 394 to 1014 $m/z$). Settings for the MS were: 60,000 resolution; 100% normalised AGC target; centroid. Settings for the DIA scans were: 30,000 resolution; 30% normalised HCD collision energy; 1000% normalised AGC target; 55 ms max. injection time; centroid.

Raw data were processed using DIA-NN (version 2.1.0 for academic research (Demichev et al, 2020)) with default settings unless stated otherwise. DIA-NN was initially used to generate a predicted spectral library, which was searched against the human UniProt FASTA file (release: 04/2025; reviewed and canonical sequences only; including the universal contaminants (Frankenfield et al, 2022)). Search parameters included: specific protease Trypsin (allowing one missed cleavage and two variable modifications); fixed modification: carbamidomethylation (C); variable modifications: Oxidation (M) and N-terminal acetylation. Quantification was performed using DIA-NN's built-in quantification strategy set

to QuantUMS high precision mode. The resulting report.pg_ma-trix.tsv file was used for downstream analysis. Contaminant entries were removed. Protein groups were retained only if they were supported by at least two prototypic peptide sequences and showed a 100% quantification rate in at least one condition. Missing values were imputed in three steps: (1) Proteins missing entirely in one condition but present in all replicates of another (missing not at random; MNAR) were imputed using the SampMin method (lowest intensity per sample); (2) if a protein had ≥50% quantification rate, group-wise mean imputation (missing at random; MAR) was applied; (3) remaining missing values were imputed using tail-based imputation. Batch effects between replicates were corrected using the ComBat algorithm implemented in the sva package (Johnson et al, 2007), with empirical Bayes disabled (par.prior = FALSE). Differential protein abundance analysis was performed using the limma package (Ritchie et al, 2015). All downstream analyses and visualisations were conducted in RStudio (version 2024.4.2.764) using R (version 4.3.3).

## Statistical analysis

All experiments were replicated at least once, with results obtained similar to the data displayed. No data were excluded from analysis. Microscopy images were initially preprocessed using the median filter function with a specified radius in Fiji (Schindelin et al, 2012). Nuclei, MAR, and p62 foci were identified either using the Find Maxima function in Fiji or segmented and detected with CellProfiler (Stirling et al, 2021). The number of foci per cell was statistically analysed by applying the non-parametric Kruskal–Wallis test, followed by pairwise comparisons using the Mann–Whitney $U$-test to compare different treatments. On average, 4–5 image frames containing 15–30 cells per frame were analysed for each experimental condition. The base R package and ggplot2 were used for statistical testing and plotting, respectively (Team, 2024; Wickham, 2009).

# Data availability

Plasmids generated by our lab are available via Addgene. Microscopy images are available on the BioImage Archive under the accession number S-BIAD2124 (https://www.ebi.ac.uk/biostudies/BioImages/studies/S-BIAD2124). The mass spectrometry proteomics data have been deposited to the ProteomeXchange Consortium via the PRIDE (Perez-Riverol et al, 2025) partner repository with the dataset identifier PXD065525 (http://www.ebi.ac.uk/pride/archive/projects/PXD065525).

The source data of this paper are collected in the following database record: biostudies:S-SCDT-10_1038-S44318-025-00545-7.

# Peer review information

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

## Acknowledgements

We thank all the members of our lab for fruitful discussions as well as Wiwik Bauten and Gülcan Aydin for technical support. Phthal01 was a gift from Michael Cohen. We are grateful to Mirna Barsoum and Christian Preisinger (both IZKF) for the mass spectrometry and bioinformatic analysis of the cell lysates. This work was supported by the Deutsche Forschungsgemeinschaft (DFG) to KLHF-Z (FE1423/3-1), by the START programme of the Faculty of Medicine, RWTH Aachen University to NJI (015/24) and by the Habilitation Stipendium of the Faculty of Medicine, RWTH Aachen University to KLHF-Z. This work was supported by the Confocal Microscopy Facility and by the Proteomics Facility, Core Facilities of the Interdisciplinary Center for Clinical Research (IZKF) Aachen within the Faculty of Medicine at RWTH Aachen University.

## Author contributions

**Nonso J Ikenga**: Resources; Formal analysis; Funding acquisition; Investigation; Writing—review and editing. **Jörg Vervoorts**: Resources; Methodology; Writing—review and editing. **Bernhard Lüscher**: Resources; Writing—review and editing. **Roko Žaja**: Conceptualisation; Resources; Formal analysis; Supervision; Investigation; Visualisation; Methodology; Writing—review and editing. **Karla L H Feijs-Žaja**: Conceptualisation; Resources; Formal analysis; Supervision; Funding acquisition; Investigation; Visualisation; Methodology; Writing—original draft; Writing—review and editing.

Source data underlying figure panels in this paper may have individual authorship assigned. Where available, figure panel/source data authorship is listed in the following database record: biostudies:S-SCDT-10_1038-S44318-025-00545-7.

## Funding

## Disclosure and competing interests statement

The authors declare no competing interests.

