## [Peer Review File · The EMBO Journal]

PARP7 is a proteotoxic stress sensor which labels proteins for degradation

Nonso Ikenga, Joerg Vervoorts, Bernhard Luscher, Roko Zaja, and Karla Feijs

Corresponding author(s): Karla Feijs (kfeijs@ukaachen.de), Roko Zaja (rzaja@ukaachen.de)

Review Timeline:

Submission Date:	14th Jan 25
Editorial Decision:	23rd Mar 25
Revision Received:	1st Jul 25
Editorial Decision:	18th Jul 25
Revision Received:	5th Aug 25
Accepted:	6th Aug 25

Editor: Hartmut Vodermaier

Transaction Report:

Dr. Karla Feijs
Institute of Biochemistry and Molecular Biology, RWTH Aachen University
Germany

23rd Mar 2025

Re: EMBOJ-2025-120138
PARP7 is a proteotoxic stress sensor which labels proteins for autophagic degradation

Dear Dr. Feijs,

Thank you for submitting your full manuscript on PARP7 as proteotoxic stress sensor study to The EMBO Journal. Three expert referees have now reviewed it and provided the comments copied below. Since they all appreciate the timeliness and potential interest of your findings, we would be happy to pursue a revised version further for publication. Nevertheless, the referees raise a number of overlapping important concerns that would need to be adequately addressed before acceptance. These include not only requests for additional controls, quantification, or improved data/image presentation, but also the strengthening of specific conclusions as requested by referee 3, and deepening the insights into certain aspects as suggested by referees 1 and 2. I realize that not all of these points may warrant additional experimentation, but since we allow only a single (major) revision round, it will still be important to carefully respond to all queries at the time of resubmission. Therefore, I would encourage you to contact me with a tentative point-by-point response and revision plan already during the early stages of your revision work, so that we could discuss how key issues raised in the reports might best be resolved. We would also be open to extending the revision deadline if that should be helpful.

Further information on preparing, formatting and uploading a revised manuscript can be found below and in our Guide to Authors. Thank you again for the opportunity to consider this work for The EMBO Journal, and I look forward to hearing from you.

With kind regards,

Hartmut

9) To facilitate reproducibility and cross-laboratory adoption of methodologies, please structure the Materials & Methods section as outlined in our guide to authors, including a completed Reagents and Tools Table that can be downloaded from our author guidelines as well (<https://www.embopress.org/page/journal/14602075/authorguide#structuredmethods>).

10) Digital image enhancement is acceptable practice, as long as it accurately represents the original data and conforms to community standards. If a figure has been subjected to significant electronic manipulation, this must be clearly noted in the figure legend and/or the 'Materials and Methods' section. The editors reserve the right to request original versions of figures and the original images that were used to assemble the figure. Finally, we generally encourage uploading of numerical as well as gel/blot image source data; for details see: embopress.org/page/journal/14602075/authorguide#sourcedata

At EMBO Press, we ask authors to provide source data for the main manuscript figures. Our source data coordinator will contact you to discuss which figure panels we would need source data for and will also provide you with helpful tips on how to upload and organize the files.

In the interest of ensuring the conceptual advance provided by the work, we recommend submitting a revision within 3 months (21st Jun 2025). Please discuss the revision progress ahead of this time with the editor if you require more time to complete the revisions. Use the link below to submit your revision:

Link Not Available

Referee #1:

The teams of Roko Zaja and Karla Feijs have submitted a manuscript reporting how PARP7 acts as a proteotoxic stress sensor that catalyzes the mono-ADP-ribosylation (MARylation) of proteins, promoting their sequestration into cytoplasmic foci containing ubiquitin and p62. These MARylated proteins are subsequently transported to aggresomes and degraded via autophagy, highlighting an intriguing role for the PARP7 enzyme in protein quality control. The manuscript makes a timely contribution to the expanding field of non-canonical members of the large mammalian ARTD family of NAD-dependent enzymes, revealing a compelling link between PARP7 protein levels, cellular stress, and proteostasis. While the manuscript presents interesting findings and the quality of the results is suitable for publication, providing further insight into the targets of MARylation/PARYlation and elucidating how ubiquitin contributes to PARP7 proteostasis would significantly enhance the scope and impact of this study.

Experimental suggestions:

- Given that a substantial portion of the validation work on PARP7 utilizes the PARP7 inhibitor RBN2397 and that concerns exist regarding its cross-reactivity with PARP2 and other ARTD-family members (see comments in Sanderson et al., Cell Chem Biol, doi.org/10.1016/j.chembiol.2022.11.012), it would be beneficial to repeat key assays using a second PARP7 inhibitor. KMR-206, a different class of PARP7 inhibitor with a distinct selectivity profile compared to RBN2397 (a first-generation PARP7i), could be a suitable candidate. Would KMR-206 also stabilize PARP7 protein levels, as RBN2397 appears to do directly or indirectly (Fig. 5c)?

- Figure 2E clearly demonstrates differences in MARylation/PARYlation patterns between HeLa and U2OS cells. Specifically, HeLa cells exhibit a predominant band at approximately 80 kDa that becomes enriched and heavily MARylated/PARYlated, whereas U2OS cells display a band at around 120 kDa. Can the authors provide an explanation for this? Are there any potential candidate proteins that could be identified and tested? Given that these proteins are distinctly enriched, and tools for detecting them are available, could they be purified using the same antibodies and identified by mass spectrometry (MS) or validated with specific antibodies (if candidate ADP-ribosylated targets are known or can be proposed)?

- Have the ubiquitination sites on PARP7 been mapped? If so, could the authors mutate one or more of these sites and examine how such modifications impact PARP7 protein levels and turnover?

Comments on figures containing imaging data:

Several figures present immunofluorescence or related imaging data that are somewhat difficult to interpret, particularly in print. Would it be possible to zoom in or adjust the contrast (without excessive manipulation of the results) to enhance visibility? The green channels for MARylation/PARYlation in Figures 2a, 2c, 2d, and 3a, 3b appear problematic. While I do not believe this issue is related to my red-green colorblindness, I would appreciate the authors addressing this concern.

In this context, Figure 4a would benefit from the inclusion of a relevant positive control for MARylation/PARYlation and ubiquitin, as the four blank black images alone are not convincing.

Additional comment:

The manuscript contains a few typographical errors, particularly where "PARP" has been inadvertently replaced with "PAPR" due to autocorrection in MS Word. This should be corrected throughout the manuscript.

Referee #2:

Review for EMBO-2025-120138

The paper by Ikenga et al. provides a detailed characterization of ADP-ribosylation during proteotoxic stress (PS) induction. While previous studies have suggested the involvement of ADP-ribosylation in PS, the specific PARPs that play a role have not been identified. The authors provide evidence that PARP7-mediated ADP-ribosylation is a sensor for PS proteotoxic stress, directing proteins toward autophagic degradation.

In conclusion, the manuscript is well-written, and the experiments are generally well-executed. These studies set the stage for further studies exploring the mechanism by which ADP-ribosylation could lead to autophagic degradation. The growing interest in ADP-ribosylation and the specific cellular functions of individual PARP family members should make this manuscript appealing to the broad readership of the EMBO journal. However, I recommend addressing the following items before publication.

1. Fig. 1A: What percentage of cells exhibit MAR/PAR cytoplasmic foci? The authors should quantify this. If it appears in only a subset of cells, it may indicate a cell-cycle dependence on this phenomenon.
2. Fig. E: The different kinetics/pattern of ADP-ribosylation induction in HeLa versus U2OS cells is intriguing. Do the authors think this reflects the different PARP family members that are present? In particular, the band between 75 and 100 kDa, which they later show is PARP7 dependent, is more prominent in HeLa cells at later time points, whereas the band ~130 kDa is more prominent in U2OS cells, especially at very early time points. The authors should discuss these differences in more detail.
3. Fig 3D: If the authors used a MAR-specific antibody, would they see an even more significant increase in "insoluble" MARylated proteins upon inhibiting both the proteasome and autophagy? This would help support the idea that MARylated proteins are being degraded by autophagy when the proteasome is inhibited.
4. Fig 3: these results would be bolstered by the use of other inhibitors of autophagy (e.g., 3-methyladenine).
5. Fig. 4A: Did the authors try alternative fixation conditions to see if the ADP-ribosylation signal is present by IF under TAK-243 conditions? Or perhaps a MAR-specific antibody?
6. Fig. 4C: Does TAK-243 induce the expression (transcriptionally or translationally) of PARP7?
7. Fig. 4C: The increase in insoluble ADP-ribosylation (following TAK-243 treatment) seems to apply only to PARP10; for other PARPs, ADP-ribosylation rises in soluble and insoluble fractions. The authors should mention this in the text.
8. Fig. 5C: Inhibition of PARP7 appears to decrease p62, and expression of PARP7 appears to increase p62 (input lanes). It might be worth quantifying these results as they could be relevant to the mechanism of action of PARP7/PARP7 inhibitor.
9. Fig. 6A,B: It is intriguing that a pan-PARPi prevents p62 foci formation. The authors should determine if a PARP7 or tankyrase inhibitor (or combined) has the same effect. This would provide more insight into the PARP regulating the formation of p62 foci.
10. Fig 6D: Are the results only significant for PARP7 knockdown? It seems that the knockdown of other PARP family members results in a milder phenotype, but it would be helpful to confirm if this is significant.

Referee #3:

In this study, Ikenda et al. found that short-term DUB inhibition leads to the formation of MAR-positive structures. Mono-ADP-Ribosylation (MAR)

proteins are recruited to the ALIS/p62 body and ultimately degraded by autophagy. They also identified Poly(ADP-ribose) polymerase 7 (PARP7) as the enzyme responsible for MAR modification and demonstrated its involvement in the ADP-Ribosylation (ADPr) of abnormal proteins in response to proteotoxic stress. The formation of foci following short-term DUB inhibition is surprising and highlights the importance of the ubiquitination cycle in autophagy. In recent years, multiple studies have linked ADP-ribosylation to the p62 body, suggesting that ADP-ribosylation could potentially serve as a novel marker for the p62 body. As the authors noted, many aspects remain to be elucidated, such as how ADPr contributes to ALIS body formation and how it promotes ubiquitination. Nonetheless, the data presented are clear. However, some additional experiments are necessary to support the conclusions.

Major Comments

1. The authors' model of ADPr and PARP7 in regulating protein homeostasis is intriguing. However, the experiments presented are inconclusive.

The authors conclude that ADPr-modified proteins are recruited to foci in a ubiquitination-dependent manner and remain diffuse in the absence of ubiquitination. However, TAK243 treatment resulted in the accumulation of MAR/PARylated proteins in the RIPA-insoluble fraction (Fig. 4B, C). This suggests the presence of ubiquitin-negative aggregates (Fig. 7C), yet MAR-positive aggregates are not observed in Fig. 4A. While the data itself is accurate, the existence of these aggregates should be experimentally demonstrated, or the conclusion should be revised accordingly.

2. Why is the PARP7 substrate not degraded by the proteasome under proteotoxic stress?

The authors claim that "ADPr is not a mark that targets proteins for proteasomal degradation, similar to K48- or K11-linked polyubiquitination, but rather is induced in response to the accumulation of proteins that cannot be degraded." If this is the case, the PARP7 substrate should not be degraded by the proteasome even under normal conditions and should accumulate following bafilomycin A1 treatment.

3. The molecular mechanism underlying the enhanced ubiquitination of MAR proteins remains unclear, as does the temporal relationship between ubiquitination and ADP-ribosylation.

Under DUB or proteasome inhibition, ubiquitination occurs first, followed by ADP-ribosylation. In contrast, TAK243 treatment induces ADP-ribosylation without the presence of ubiquitinated substrates. The relationship between these two modifications is unclear and should be explained in greater detail.

Minor Comments

1. The figure and legend for Fig. 2F do not match. Please verify.

(It does not appear to be a time-course experiment.)

2. The formation of MAR-positive foci following short-term DUB inhibition is an interesting phenomenon.

Some reports suggest that high concentrations of PR619 induce ER stress. To eliminate potential side effects, a dose-dependency analysis or alternative DUB inhibitors (e.g., NEM) should be considered.

3. Inhibitor experiments in Fig. 4D show that PARP7 and TNKS1 inhibitors have nearly identical effects.

What is the effect of TNKS on foci formation? Similar to PARP7 knockdown (Fig. 6C), does TNKS knockdown inhibit DUBi-dependent foci formation?

4. Recent reports indicate that p62 undergoes PARP14-dependent ADP-ribosylation (Kubon et al., 2024).

Does PARP7 also ADP-ribosylate p62, and does this modification affect foci formation?

5. The statistical analysis for Fig. 6D is missing.

Please provide the relevant statistical data.

6. The term "p62 body" is more commonly used than "ALIS body" and may be more widely understood by readers.

We are very grateful to all three reviewers for the careful assessment of our work. Although all reviewers are in general positive, several suggestions were made. We have integrated a quantitative proteomics experiment which has revealed some of PARP7's potential substrates to us, even though measured effects were small. Lastly, as in the meantime several preprints have appeared which have direct influence on our work, we have updated our model to include some of their findings.

Referee #1:

The teams of Roko Zaja and Karla Feijs have submitted a manuscript reporting how PARP7 acts as a proteotoxic stress sensor that catalyzes the mono-ADP-ribosylation (MARylation) of proteins, promoting their sequestration into cytoplasmic foci containing ubiquitin and p62. These MARylated proteins are subsequently transported to aggresomes and degraded via autophagy, highlighting an intriguing role for the PARP7 enzyme in protein quality control. The manuscript makes a timely contribution to the expanding field of non-canonical members of the large mammalian ARTD family of NAD-dependent enzymes, revealing a compelling link between PARP7 protein levels, cellular stress, and proteostasis. While the manuscript presents interesting findings and the quality of the results is suitable for publication, providing further insight into the targets of MARylation/PARYlation and elucidating how ubiquitin contributes to PARP7 proteostasis would significantly enhance the scope and impact of this study.

Experimental suggestions:

- Given that a substantial portion of the validation work on PARP7 utilizes the PARP7 inhibitor RBN2397 and that concerns exist regarding its cross-reactivity with PARP2 and other ARTD-family members (see comments in Sanderson et al., Cell Chem Biol, doi.org/10.1016/j.chembiol.2022.11.012), it would be beneficial to repeat key assays using a second PARP7 inhibitor. KMR-206, a different class of PARP7 inhibitor with a distinct selectivity profile compared to RBN2397 (a first-generation PARP7i), could be a suitable candidate. Would KMR-206 also stabilize PARP7 protein levels, as RBN2397 appears to do directly or indirectly (Fig. 5c)?

This is a valuable suggestion to confirm the results obtained with RBN2397 with a more selective inhibitor. In new **Supplementary Figure 1**, we included an experiment performed with cells incubated with KMR-206. Here we see that also this compound stabilizes PARP7, indicating that this is not an off-target effect.

- Figure 2E clearly demonstrates differences in MARylation/PARYlation patterns between HeLa and U2OS cells. Specifically, HeLa cells exhibit a predominant band at approximately 80 kDa that becomes enriched and heavily MARylated/PARYlated, whereas U2OS cells display a band at around 120 kDa. Can the authors provide an explanation for this? Are there any potential candidate proteins that could be identified and tested? Given that these proteins are distinctly enriched, and tools for detecting them are available, could they be purified using the same antibodies and identified by mass spectrometry (MS) or validated with specific antibodies (if candidate ADP-ribosylated targets are known or can be proposed)?

We have included a mass spectrometry-based experiment performed using HeLa cells treated with PARP7 inhibitor, where we asked how the inhibitor influences protein stability. Effects on protein stability are in general very mild and with a cutoff of log₂ fold change >0.5, we find approximately 100

proteins which are upregulated following PARP7 inhibition (**New Supplementary Figure 2**). This low effect is possibly due to lack of additional stressor. Even though this experiment aimed to identify those proteins whose stability is influenced by PARP7 inhibition, we reasoned that the proteins which are affected are those that are modified by PARP7. We have performed an enrichment of ADP-ribosylated proteins and detected specific proteins, including FOS as one of the proteins stabilised by PARP7 (**New Figure 5D/E**). Here, we see that following TAK243 treatment we can enrich specific proteins, which is decreased by a PARP7 inhibitor. The measured effects are not large, but this might be caused by constant turnover of substrate proteins. As to the identify of the strongest bands we cannot be fully certain, however, we have also extended the text to highlight these prominent bands and their possible identity:

“The most prominent signal in HeLa cells is present at around 75kDa, which is also observed in U2OS albeit with lesser intensity and not responding to the inhibitor treatment. In U2OS cells the most prominent ADP-ribosylation induced by proteasome inhibition is visible at a larger size. As PARP7 and PARP1 are approximately 75kD and 115kD respectively, it is possible that the strongest signals reflect automodified PARPs. This could imply that different transferases or hydrolases are active in these cells, in line with for example differences in PARG activity levels we have observed previously between HEK293 and HeLa cells”

- Have the ubiquitination sites on PARP7 been mapped? If so, could the authors mutate one or more of these sites and examine how such modifications impact PARP7 protein levels and turnover?

Unfortunately, as far as we know modification sites on PARP7 are not yet mapped. Data from other labs suggests the presence of multiple MARylation sites on PARP7 on diverse amino acids. Moreover, high-throughput studies have identified ubiquitination on PARP7 (7 lysines modified according to PhosphoSite Plus), although none of these have been verified. In an ideal scenario where both ADP-ribosylation and ubiquitination sites are known, we could work with specific mutants to decipher how the modifications contribute to protein turnover. The complexity of modifications seems too large to be easily addressed. Thus, we feel that a more in-depth analysis of PARP7 turnover, with corresponding single and combinatorial mutants, will be an important topic for future studies.

Comments on figures containing imaging data:

Several figures present immunofluorescence or related imaging data that are somewhat difficult to interpret, particularly in print. Would it be possible to zoom in or adjust the contrast (without excessive manipulation of the results) to enhance visibility? The green channels for MARylation/PARylation in Figures 2a, 2c, 2d, and 3a, 3b appear problematic. While I do not believe this issue is related to my red-green colorblindness, I would appreciate the authors addressing this concern.

In this context, Figure 4a would benefit from the inclusion of a relevant positive control for MARylation/PARylation and ubiquitin, as the four blank black images alone are not convincing.

We agree that the signals are not always easy to detect. We had chosen to display fields with several cells, to also show we did not pick individual cells for analysis, however, this is of course a trade off in terms of visibility of signals. We have now included enlargements of areas indicated in the original images, where the signals are more easily visible. The original images we will upload to an appropriate server as per the Journals instructions, so that it will always be possible to download the originals and for example further enhance contrast. We have also included a control for 4A (DUBi treated cells), where both the ubiquitin and ADPr signals are visible.

Additional comment:

The manuscript contains a few typographical errors, particularly where "PARP" has been inadvertently replaced with "PAPR" due to autocorrection in MS Word. This should be corrected throughout the manuscript.

We are grateful that the reviewer spotted these typos and have corrected this throughout the manuscript.

Referee #2:

Review for EMBO-2025-120138

The paper by Ikenga et al. provides a detailed characterization of ADP-ribosylation during proteotoxic stress (PS) induction. While previous studies have suggested the involvement of ADP-ribosylation in PS, the specific PARPs that play a role have not been identified. The authors provide evidence that PARP7-mediated ADP-ribosylation is a sensor for PS proteotoxic stress, directing proteins toward autophagic degradation.

In conclusion, the manuscript is well-written, and the experiments are generally well-executed. These studies set the stage for further studies exploring the mechanism by which ADP-ribosylation could lead to autophagic degradation. The growing interest in ADP-ribosylation and the specific cellular functions of individual PARP family members should make this manuscript appealing to the broad readership of the EMBO journal. However, I recommend addressing the following items before publication.

1. Fig. 1A: What percentage of cells exhibit MAR/PAR cytoplasmic foci? The authors should quantify this. If it appears in only a subset of cells, it may indicate a cell-cycle dependence on this phenomenon.

This is an interesting suggestion. It appeared to us that the foci are present in the vast majority of cells and thus unlikely to be cell-cycle dependent. We have now determined the number of cells and have averaged all conditions (glyoxal, methanol, ADPr or PAR/MAR antibody), where we find that 65% of DUBi-treated cells display foci. This would imply that foci form following DUBi in G1, as the largest percentage of cells will be in G1, and possibly also in S- and G2/M phases.

2. Fig. E: The different kinetics/pattern of ADP-ribosylation induction in HeLa versus U2OS cells is intriguing. Do the authors think this reflects the different PARP family members that are present? In particular, the band between 75 and 100 kDa, which they later show is PARP7 dependent, is more prominent in HeLa cells at later time points, whereas the band ~130 kDa is more prominent in U2OS cells, especially at very early time points. The authors should discuss these differences in more detail.

The reviewer correctly noticed differences between cell lines. Previously, we also noticed significant differences between HeLa and HEK293 cells in terms of RNA ADP-ribosylation and have speculated also there that this is due to a difference in transferase/hydrolase expression (Weixler et al, Communications Biology 2025). The proteins with apparent molecular weights as pointed out by the reviewer could for example correspond to PARP7 (75 kDa) and PARP1 (120 kDa). We have extended the discussion to hypothesize about the differences between U2OS and HeLa, and have now also included a mass spectrometry experiment quantifying relative protein abundance following PARP7 inhibition. In supplementary Figure 2B, we have included a table with the most upregulated proteins with as top hit PARP7. We have adjusted the text as follows:

“The most prominent signal in HeLa cells is present at around 75kDa, which is also observed in U2OS albeit with lesser intensity and not responding to the inhibitor treatment. In U2OS cells the most prominent ADP-ribosylation induced by proteasome inhibition is visible at a larger size. As PARP7 and PARP1 are approximately 75kD and 115kD respectively, it is possible that the strongest signals reflect automodified PARPs. This could imply that different transferases or hydrolases are active in these cells, in line with for example differences in PARG activity levels we have observed previously between HEK293 and HeLa cells”

3. Fig 3D: If the authors used a MAR-specific antibody, would they see an even more significant increase in "insoluble" MARylated proteins upon inhibiting both the proteasome and autophagy? This would help support the idea that MARylated proteins are being degraded by autophagy when the proteasome is inhibited.

4. Fig 3: these results would be bolstered by the use of other inhibitors of autophagy (e.g., 3-methyladenine).

We have probed blots using a MAR-specific antibody and although we do observe similar effects compared to the MAR/PAR antibody, the induction of signal following proteasome inhibition is not more pronounced with the MAR antibody (included in **new Figure 3E**). We generated the majority of data in the manuscript using the MAR/PAR antibody, as we have previously observed that this antibody has the broadest substrate spectrum (Weixler et al, Life Science Alliance 2023). The MAR-specific antibodies were designed to recognise the ADPr only in the context of a specific amino acid linkage (serine or arginine) and/or peptide surrounding. We cannot be certain that this antibody detects all ADP-ribosylation occurring on cysteine as well. If the reagents would recognise ADPr independent of amino acid linkage or backbone, then presumably they would be able to detect PAR as well, considering how this consists of several ADPr moieties linked together. Key experiments were done with a MAR-antibody to ensure that we observe MARylation in these experiments (Fig 1C/D, 3E, Supplementary Figure 3).

We furthermore agree with the reviewer that robustness of our data can be enhanced by including an additional inhibitor of autophagy. We have performed experiments using siRNA for Atg5 and Atg7, as these are essential for autophagy and their knockdown will prevent formation of the autophagosome (**New Figure 3E**). Also here we see an increase in ADP-ribosylation, confirming the results obtained with bafilomycin A.

5. Fig. 4A: Did the authors try alternative fixation conditions to see if the ADP-ribosylation signal is present by IF under TAK-243 conditions? Or perhaps a MAR-specific antibody?

The reviewer is correct to suggest trying different fixation and staining methods, as it is a little puzzling that we observe a substantial increase in ADPr signal on western blots but not when using microscopy. We have previously observed that fixation using PFA and staining with the PAR/MAR antibody leads to staining of mitochondria, with presumably metabolites being detected (Weixler et al, Life Science Alliance 2023). LC3 foci are undetectable when cells are permeabilised with Triton-X100, but will remain when digitonin is used instead. We have attempted different fixatives, used both Triton-X100 or digitonin as detergents and have stained the slides with both PAR/MAR as well as ADPr antibodies. We still cannot observe an ADPr signal in those cells. Several explanations are possible: the ADPr in this condition may be obscured by proteins binding to it, or may be present in larger structures inaccessible to antibodies, or due to lack of denaturing as in western blot remain undetected. We have altered the text to reflect this negative result better, as we cannot conclude where the ADPr-signal as observed in western blot resides in the cells (Figure 7C was changed to reflect this).

6. Fig. 4C: Does TAK-243 induce the expression (transcriptionally or translationally) of PARP7?

Unfortunately, no good PARP7 antibodies are available to our knowledge – we tested the antibody from ThermoFischer (PA5-40774) but could not detect PARP7. We have performed RT-qPCR of PARP7 mRNA levels in response to treatments with BTZ, PARP7 inhibitor and TAK243. We decided not to include the result in the manuscript, as all the housekeeping genes we tested responded to TAK243 treatment. When we analyse the results using the $\Delta\Delta C_t$ method anyway, it appears that TAK243 reduces PARP7 mRNA levels, however, due to the measured effects on housekeeping genes we cannot be sure this effect is genuine.

7. Fig. 4C: The increase in insoluble ADP-ribosylation (following TAK-243 treatment) seems to apply only to PARP10; for other PARPs, ADP-ribosylation rises in soluble and insoluble fractions. The authors should mention this in the text.

The reviewer is correct with this observation, and we have adjusted the text accordingly. We have also expanded the text to refer to a recent publication which has shown that ADP-ribose on PARP10 can be further modified with ubiquitin (<https://doi.org/10.1038/s44318-025-00391-7>). Here, the authors observed a similar “smeared” signal for PARP10, which can be collapsed into a single band by treatment with DUBs. Presumably, we observe the same dual modification which is lost upon ubiquitination.

8. Fig. 5C: Inhibition of PARP7 appears to decrease p62, and expression of PARP7 appears to increase p62 (input lanes). It might be worth quantifying these results as they could be relevant to the mechanism of action of PARP7/PARP7 inhibitor.

The reviewer is right that in the mentioned experiment it seemed like there was an effect on p62 levels. We did however not find any consistent effect when comparing diverse datasets. We have also looked at the mass spectrometry dataset, where SQSTM1 was found to be slightly less abundant in the PARP7 inhibitor treated sample, but with an effect of only 1.2 fold difference (log₂FC of - 0.35). Also in line with reviewer 3s comments, we have added a short discussion about a potential role for p62 ADP-ribosylation in this process:

“It is possible that p62 itself forms a relevant ADP-ribosylation substrate in this context, as it was reported to be modified by PARP14 (Kubon *et al.*, 2024). In our mass spectrometry dataset p62 stability was only slightly different by PARP7 inhibition (± 1.2 fold downregulated), nevertheless p62 could form a direct PARP7 substrate following stress.”

9. Fig. 6A,B: It is intriguing that a pan-PARPi prevents p62 foci formation. The authors should determine if a PARP7 or tankyrase inhibitor (or combined) has the same effect. This would provide more insight into the PARP regulating the formation of p62 foci.

We agree with the reviewer that it is important, which is why we continued with the experiments in 6C. Here we probed siRNAs for individual PARPs, and found that PARP7 is required for foci formation. We have now performed these experiments with PARP7i or TNKSi and find that also PARP7i reduces formation, similar to PARP7 siRNA, whereas the TNKS1 inhibitor did not reduce foci formation (Supplementary Figure 3). This experiment confirms that PARP7 is key in this process, and that PARP7 and TNKS1 functions are clearly distinct.

10. Fig 6D: Are the results only significant for PARP7 knockdown? It seems that the knockdown of other PARP family members results in a milder phenotype, but it would be helpful to confirm if this is significant.

The reviewer correctly observed that knockdown of the other PARPs also leads to a slight reduction in foci. There are two possible explanations for this phenomenon: either the ADP-ribosylation activity of all enzymes contributes to foci formation and has some effect when lost, even if not as extensive as the loss of PARP7, or due to the large sample size small differences become statistically significant, where it is unsure whether this carries biological relevance. We have included the analysis, so that it is clear to future readers that also the other PARPs may contribute to foci formation.

Referee #3:

In this study, Ikenda *et al.* found that short-term DUB inhibition leads to the formation of MAR-positive structures. Mono-ADP-Ribosylation (MAR) proteins are recruited to the ALIS/p62 body and ultimately degraded by autophagy. They also identified Poly(ADP-ribose) polymerase 7 (PARP7) as the enzyme responsible for MAR modification and demonstrated its involvement in the ADP-Ribosylation (ADPr) of abnormal proteins in response to proteotoxic stress. The formation of foci following short-term DUB inhibition is surprising and highlights the importance of the ubiquitination cycle in autophagy. In recent years, multiple studies have linked ADP-ribosylation to the p62 body, suggesting that ADP-ribosylation could potentially serve as a novel marker for the p62 body. As the authors noted, many aspects remain to be elucidated, such as how ADPr contributes to ALIS body formation and how it

promotes ubiquitination. Nonetheless, the data presented are clear. However, some additional experiments are necessary to support the conclusions.

Major Comments

1. The authors' model of ADPr and PARP7 in regulating protein homeostasis is intriguing. However, the experiments presented are inconclusive.

The authors conclude that ADPr-modified proteins are recruited to foci in a ubiquitination-dependent manner and remain diffuse in the absence of ubiquitination. However, TAK243 treatment resulted in the accumulation of MAR/PARYlated proteins in the RIPA-insoluble fraction (Fig. 4B, C). This suggests the presence of ubiquitin-negative aggregates (Fig. 7C), yet MAR-positive aggregates are not observed in Fig. 4A. While the data itself is accurate, the existence of these aggregates should be experimentally demonstrated, or the conclusion should be revised accordingly.

We agree that at present we do not have firm evidence for the presence of ADPr-positive aggregates. We have attempted additional fixation and permeabilisation methods to visualise these aggregates (in line with reviewer 2 comment 5), but were unable to visualise ADP-ribosylation following TAK243 in HeLa cells. Of note, in earlier works discrepancies between immunofluorescence stainings of ADPr and western blots were also observed, with the increase in ADP-ribosylation detected using western blots not reflected in immunofluorescence. It is possible that these aggregates may exist but not be accessible using our immunofluorescence methods, or indeed that the ADP-ribosylated proteins in this condition are dispersed and therefore difficult to detect. We have adjusted the scheme in Fig. 7C as well as the text correspondingly.

2. Why is the PARP7 substrate not degraded by the proteasome under proteotoxic stress?

The authors claim that "ADPr is not a mark that targets proteins for proteasomal degradation, similar to K48- or K11-linked polyubiquitination, but rather is induced in response to the accumulation of proteins that cannot be degraded." If this is the case, the PARP7 substrate should not be degraded by the proteasome even under normal conditions and should accumulate following bafilomycin A1 treatment.

The reviewer correctly points out a flaw in our logic. The fact that ADP-ribosylation occurs later than ubiquitination does not mean that it cannot lead to proteasomal degradation. It might show that PARPs are stabilised following stress, which then label proteins for degradation via either proteasome or autophagy. As bafilomycin A1 does lead to an increase in ADP-ribosylated proteins (3D; 3E; 6E), there is evidence for processing of ADPr-proteins via autophagy. We can however not exclude that also proteasomal degradation occurs. We have altered the manuscript to ensure correct interpretation of the data and have also slightly altered the title to reflect this. The title now reads: **PARP7 is a proteotoxic stress sensor which labels proteins for degradation**, where the word "autophagic" was removed. We have also extended the discussion to make it clear that this work has raised several questions which have to be addressed in future studies, including a more detailed assessment of the different degradation machineries involved.

3. The molecular mechanism underlying the enhanced ubiquitination of MAR proteins remains unclear, as does the temporal relationship between ubiquitination and ADP-ribosylation.

Under DUB or proteasome inhibition, ubiquitination occurs first, followed by ADP-ribosylation. In contrast, TAK243 treatment induces ADP-ribosylation without the presence of ubiquitinated substrates. The relationship between these two modifications is unclear and should be explained in greater detail.

This is a very important question. In our working model, DUB and proteasome inhibition lead to blockage of proteasomal degradation with as consequence stabilisation of PARP7. The ADP-ribosylation generated by the stabilised PARP7 does not need to occur on the polyubiquitinated proteins but can occur independent thereof on different proteins. Those ADP-ribosylated proteins need ubiquitination for further processing, as in absence of ubiquitination ADP-ribosylated proteins accumulate (TAK243 experiments).

DTX2 has been described to be recruited to some PARP7 substrates (for example to the androgen receptor) and ubiquitinate those targets to induce degradation. Intriguingly recent work showed that ADP-ribose can be modified with ubiquitin in cells. This modification, termed MARUbylation, is introduced by DTX2 on automodified PARP10 (<https://doi.org/10.1038/s44318-025-00391-7>). Two further preprints have appeared, showing that ADP-ribosylated PARP7 is modified by DTX2 to generate the MARUbylation. This dual ADPr-ubiquitin, MARUbe, is then recognised by RNF114 and further modified with polyubiquitin (<https://doi.org/10.1101/2025.05.11.653360> and <https://doi.org/10.1101/2025.05.08.652854>). Although these papers have not addressed the fate of MARUbylated proteins, it is possible that MARUbylation plays a role in degradation. Considering that these recent preprints are dedicated to investigating the further modification of ADP-ribose with ubiquitin, we believe it is beyond the scope of the current work to pinpoint the E3 ligase responsible. To inform future readers of our work that it is possible to add ubiquitin onto ADP-ribose, we have added this possibility to our scheme in 7A. As we have not tested this possibility, we have clearly indicated the hypothetical nature of this model with question marks.

Minor Comments

1. The figure and legend for Fig. 2F do not match. Please verify.
(It does not appear to be a time-course experiment.)

We apologise for this mistake which the reviewer spotted well. The legends for 2E and 2F were swapped, presumably due to reworking the manuscript and figures. We have now corrected the legend.

2. The formation of MAR-positive foci following short-term DUB inhibition is an interesting phenomenon.

Some reports suggest that high concentrations of PR619 induce ER stress. To eliminate potential side effects, a dose-dependency analysis or alternative DUB inhibitors (e.g., NEM) should be considered.

The reviewer is justified to inquire about potential side effects of DUBi. We have now treated cells with NEM to determine whether it has effects comparable to PR-619. The ADP-ribosylation induced by NEM leads to a strong nuclear signal. NEM can alkylate cysteine residues and therefore have off-target effects, which may be what we observe here. Therefore, we also tested 1 μ M PR-619, which at an incubation time of 30 minutes is sufficient to induce ADPr foci. At this concentration and treatment duration, it is unlikely that significant ER stress occurs although it can still not be excluded. Therefore, we then also treated cells with thapsigargin at 500nM and 1 μ M to induce ER stress, however, also this did not lead to detectable foci. We have not included these experiments with the current version of the manuscript but can upload these results as supplementary data if required.

3. Inhibitor experiments in Fig. 4D show that PARP7 and TNKS1 inhibitors have nearly identical effects.

What is the effect of TNKS on foci formation? Similar to PARP7 knockdown (Fig. 6C), does TNKS knockdown inhibit DUBi-dependent foci formation?

This is a very good question, which we have now addressed with the usage of TNKS1/2 inhibitor XA939. When we add a PARP7 inhibitor foci formation is reduced, whereas the TNKS1/2 inhibitor XAV939 has no effect or even slightly increases foci formation (Supplementary Figure 3). These results indicate that the foci we observe depend on PARP7, but not TNKS activity, underscoring the different roles of PARP7 and TNKS in this process.

4. Recent reports indicate that p62 undergoes PARP14-dependent ADP-ribosylation (Kubon et al., 2024).

Does PARP7 also ADP-ribosylate p62, and does this modification affect foci formation?

We have immunoprecipitated ADP-ribosylated proteins and probed presence of p62 in this IP similar to the experiment in Figure 5E, however, were unable to detect p62. Testing whether p62 modification affects foci formation is unfortunately beyond the scope of the current work, as this would involve mapping of modification sites to be able to mutate and analyse them. Mapping of ADP-ribosylation sites is not straightforward and needs expert analysis, which we are unable to do, but which could be part of a future collaborative project studying the interplay between PARP7 (and PARP14) and p62. We have expanded the discussion to include this important aspect in our manuscript with the following text:

“It is possible that p62 itself forms a relevant ADP-ribosylation substrate in this context, as it was reported to be modified by PARP14 (Kubon *et al.*, 2024). In our mass spectrometry dataset p62 stability was only slightly different by PARP7 inhibition (± 1.2 fold downregulated), nevertheless p62 could form a direct PARP7 substrate following stress.”

5. The statistical analysis for Fig. 6D is missing.

Please provide the relevant statistical data.

We apologise for this omission and have now added the analysis.

6. The term "p62 body" is more commonly used than "ALIS body" and may be more widely understood by readers.

We appreciate this comment and have adjusted the terminology used throughout the paper.

Dr. Karla Feijs
RWTH Aachen University
Institute of Biochemistry and Molecular Biology, RWTH Aachen University
Aachen
Germany

18th Jul 2025

Re: EMBOJ-2025-120138R
PARP7 is a proteotoxic stress sensor which labels proteins for degradation

Dear Karla,

Thank you for submitting your revised manuscript for our consideration. It has now been assessed once more by the original referees 2 and 3, whose comments are copied below. While referee 2 only retains a minor presentational issue that I would leave to your discretion, referee 3 asks for biochemical and/or cell-biological validation of proteomically identified candidate substrates. I realize that any such data you may already have would certainly strengthen the study, but given that this is a new request and that the proteomics analysis had been added mainly in response to referee 1, I feel that it would also be sufficient to respond only in writing to this remaining concern of referee 3.

Before we shall be able to proceed with acceptance of the study, there are however still several editorial points to be taken care of:

- Most importantly, our routine pre-acceptance image checks brought up several issues that would need to be satisfactorily clarified:
 - 1) One of the panels in Appendix Figure S3A appears to be empty/without any signals. Please check and clarify, and also provide accompanying source data for this figure panel.
 - 2) There is apparent incongruence between Figure 5C and the corresponding source data. Some of the blots presented in the figure contain empty space devoid of any (background) signal, which is inconsistent with the included source data. Furthermore, labeling of the source data is insufficient to understand in which ways it has been employed to generate the presented figure, and whether the lanes presented together really belong to the same experiment or not. This should be clarified through clear labeling of all corresponding source data and, if applicable, revision of the figure presentation.
- Please carefully go through the reference list and make sure that each reference is complete with citation year, volume, and page/locator numbers. Furthermore, also database citations such as the protocols.io references require a unique identifier, in such as a DOI or direct access link.
- Remaining text modifications needed are the deletion of the reviewer access links in the Data Availability section, removal of Appendix Figure legends (which should only be in the Appendix itself), and renaming of materials and methods to just Methods.
- Also, as we are switching from a free-text author contribution statement towards a more formal statement based on Contributor Role Taxonomy (CRediT) terms, please remove the present Author Contribution section and instead specify each author's contribution(s) directly in the Author Information page of our submission system during upload of the final manuscript. See <https://casrai.org/credit/> for more information.
- Finally, please provide suggestions for a short 'blurb' text prefacing and summing up the study in two sentences (max. 250 characters), followed by 3-5 one-sentence 'bullet points' with brief factual statements of key results of the paper; they will form the basis of an editor-written 'Synopsis' accompanying the online version of the article. Please also upload a synopsis image, which can be used as a "visual title" for the synopsis section of your paper. The image should be in PNG or JPG format with the modest dimensions of EXACTLY 550 pixels wide and 300-600 pixels high (i.e., should be much simpler/high-level than the summary figure 7).

I am therefore returning the manuscript to you for a final round of revision, hoping you will be able to address the remaining referee and editorial points in a straightforward manner. Please do not hesitate to contact me should you have any questions in this regard!

With kind regards,

Hartmut

*** PLEASE NOTE: All revised manuscripts are subject to initial checks for completeness and adherence to our formatting guidelines. Revisions may be returned to the authors and delayed in their editorial re-evaluation if they fail to comply to the following requirements (see also our Guide to Authors for further information):

9) To facilitate reproducibility and cross-laboratory adoption of methodologies, please structure the Materials & Methods section as outlined in our guide to authors, including a completed Reagents and Tools Table that can be downloaded from our author guidelines as well (<https://www.embopress.org/page/journal/14602075/authorguide#structuredmethods>).

10) Digital image enhancement is acceptable practice, as long as it accurately represents the original data and conforms to community standards. If a figure has been subjected to significant electronic manipulation, this must be clearly noted in the figure legend and/or the 'Materials and Methods' section. The editors reserve the right to request original versions of figures and the original images that were used to assemble the figure. Finally, we generally encourage uploading of numerical as well as gel/blot image source data; for details see: embopress.org/page/journal/14602075/authorguide#sourcedata

In the interest of ensuring the conceptual advance provided by the work, we recommend submitting a revision within 3 months (16th Oct 2025). Please discuss the revision progress ahead of this time with the editor if you require more time to complete the

revisions. Use the link below to submit your revision:

Link Not Available

Referee #2:

The authors have thoroughly addressed my comments and concerns. My one suggestion is that the data in Fig. S3 be moved into Fig. 6 as it further supports the idea that PARP7 is the major PARP responsible for MAR foci formation.

Referee #3:

In the revised manuscript, Ikenda et al. have carefully re-evaluated their data and addressed several of the initial concerns. They have revised the schematic representations and added explanations that are better grounded in the data. Furthermore, the inclusion of proteomic analysis provides interesting insights and suggests potential PARP7 substrates. However, the authors do not provide direct evidence of ADP-ribosylation of the candidate substrates, nor do they demonstrate their localization to p62-positive condensates upon DUB inhibition. Additional biochemical and cell biological validation is necessary to substantiate these claims.

Major Comments:

1. The authors performed quantitative proteomic analysis to enhance the reliability of their findings. These data are indeed intriguing and suggest a broader role for PARP7 in proteostasis. Among the putative substrates, the authors highlight c-Fos as a potential target of PARP7. However, there are no localization or biochemical data to support this. For example, does c-Fos translocate to MAR-, p62-, and ubiquitin-positive condensates upon DUB inhibition? Does it accumulate in the RIPA-insoluble fraction upon E1 inhibition? Such data would significantly strengthen the authors' conclusion that c-Fos is a bona fide PARP7 substrate and relevant to the proposed model.

In the revised manuscript, Ikenda et al. have carefully re-evaluated their data and addressed several of the initial concerns. They have revised the schematic representations and added explanations that are better grounded in the data. Furthermore, the inclusion of proteomic analysis provides interesting insights and suggests potential PARP7 substrates. However, the authors do not provide direct evidence of ADP-ribosylation of the candidate substrates, nor do they demonstrate their localization to p62-positive condensates upon DUB inhibition. Additional biochemical and cell biological validation is necessary to substantiate these claims.

Major Comments:

1. The authors performed quantitative proteomic analysis to enhance the reliability of their findings. These data are indeed intriguing and suggest a broader role for PARP7 in proteostasis. Among the putative substrates, the authors highlight c-Fos as a potential target of PARP7. However, there are no localization or biochemical data to support this. For example, does c-Fos translocate to MAR-, p62-, and ubiquitin-positive condensates upon DUB inhibition? Does it accumulate in the RIPA-insoluble fraction upon E1 inhibition? Such data would significantly strengthen the authors' conclusion that c-Fos is a bona fide PARP7 substrate and relevant to the proposed model.

We are glad that the reviewer agrees with changes made to the schemes as well as description of our data. We fully agree that further investigations of potential PARP7 substrates are highly relevant. As such, this manuscript forms the starting point for a new PhD student in our lab, where we are planning to investigate PARP7 substrates in more depth, as well as further study the interplay between ubiquitin and ADPr. Although highly interesting, these investigations are unfortunately beyond the scope of the current manuscript.